# Integrated 3d flow-based multi-atlas brain structure segmentation

**Yeshu Li[2], Ziming Qiu[3], Xingyu Fan[4], Xianglong Liu[2], Eric I-Chao Chang[5], Yan Xu[1,5]***

**1** School of Biological Science and Medical Engineering, State Key Laboratory of Software Development Environment, Key Laboratory of Biomechanics, Mechanobiology of Ministry of Education and Beijing Advanced Innovation Centre for Biomedical Engineering, Beihang University, Beijing, China, **2** School of Computer Science and Engineering, Beihang University, Beijing, China, **3** Electrical and Computer Engineering, Tandon School of Engineering, New York University, Brooklyn, NY, United States of America, **4** Bioengineering College, Chongqing University, Chongqing, China, **5** Microsoft Research, Beijing, China

* xuyan04@gmail.com

**Data Availability Statement:** Our code is publicly available at https://github.com/DanielLeee/intflow. The ADNI standardized dataset is available at http://adni.loni.usc.edu/methods/mri-tool/standardized-mri-data-sets/. The MICCAI 2012 dataset is

## Abstract

MRI brain structure segmentation plays an important role in neuroimaging studies. Existing methods either spend much CPU time, require considerable annotated data, or fail in segmenting volumes with large deformation. In this paper, we develop a novel multi-atlas-based algorithm for 3D MRI brain structure segmentation. It consists of three modules: registration, atlas selection and label fusion. Both registration and label fusion leverage an integrated flow based on grayscale and SIFT features. We introduce an effective and efficient strategy for atlas selection by employing the accompanying energy generated in the registration step. A 3D sequential belief propagation method and a 3D coarse-to-fine flow matching approach are developed in both registration and label fusion modules. The proposed method is evaluated on five public datasets. The results show that it has the best performance in almost all the settings compared to competitive methods such as ANTs, Elastix, Learning to Rank and Joint Label Fusion. Moreover, our registration method is more than 7 times as efficient as that of ANTs SyN, while our label transfer method is 18 times faster than Joint Label Fusion in CPU time. The results on the ADNI dataset demonstrate that our method is applicable to image pairs that require a significant transformation in registration. The performance on a composite dataset suggests that our method succeeds in a cross-modality manner. The results of this study show that the integrated 3D flow-based method is effective and efficient for brain structure segmentation. It also demonstrates the power of SIFT features, multi-atlas segmentation and classical machine learning algorithms for a medical image analysis task. The experimental results on public datasets show the proposed method's potential for general applicability in various brain structures and settings.

## Introduction

Accurate parcellation of neural regions in human brains plays an important role in brain disorder diagnosis [1], progression assessment [2], surgical planning [3] and large-scale neuroimaging studies [4, 5]. For instance, the hippocampus, as a distinctive neural structure situated in

available at https://my.vanderbilt.edu/masi/
workshops/. The LPBA40 dataset can be found at
http://resource.loni.usc.edu/resources/atlases-
downloads/. The Hammers atlases can be obtained
from https://brain-development.org/brain-atlases/.
The IBSR dataset is downloadable at https://www.
nitrc.org/projects/ibsr. The EADC-ADNI HarP
dataset is available at http://www.hippocampal-
protocol.net/SOPs/index.php.

**Funding:** YX was supported by the National Natural
Science Foundation in China under Grant
62022010, 81771910, the Fundamental Research
Funds for the Central Universities of China from the
State Key Laboratory of Software Development
Environment in Beihang University in China, the
111 Project in China under Grant B13003, the high
performance computing (HPC) resources at
Beihang University. The funder provided support in
the form of salaries for authors YX, but did not
have any additional role in the study design, data
collection and analysis, decision to publish, or
preparation of the manuscript. The specific roles of
these authors are articulated in the 'author
contributions' section.

**Competing interests:** All authors declare that YX
had financial support from the National Natural
Science Foundation in China, and the State Key
Laboratory of Software Development Environment
in Beihang University in China; no financial
relationships with any organizations that might
have an interest in the submitted work in the
previous three years; no other relationships or
activities that could appear to have influenced the
submitted work. All funding affiliations do not alter
our adherence to PLOS ONE policies on sharing
data and materials.

the medial temporal lobe under the cerebral cortex, is crucial to memory, navigation and learning [6–8]. Its atrophy is an important clinical indicator of many brain diseases, such as Alzheimer's disease [9], schizophrenia [10] and temporal lobe epilepsy [11]. The dopaminergic dysfunction of the caudate nucleus has been shown to be related to cognition in Parkinson's disease [12, 13]. Magnetic resonance imaging (MRI) images are popular and a major focus in the study of neuroimaging, especially the accurate delineation of brain structures, which can provide neuroscientists with volumetric and structural information. Therefore, accurate and reliable brain structure segmentation in MRI data can help neuroscientists track and understand relevant disease progression better and faster.

Manual segmentation labeled by clinical pathologists has long been considered the gold standard or ground truth for neuroanatomical images. It is a laborious and time-consuming task for experts to manually delineate brain structures. The fact that there are a growing number of datasets makes labeling even more difficult. Moreover, manual segmentation is required to follow some protocols, which still need improved unity and consistency [14–18]. Thus, proposing efficient automated brain image segmentation methods will ease the burden of labeling brain anatomical regions in MRI images and speed up the labeling processes. Fully automated segmentation methods, which delineate structure boundaries directly [19–24] without any human involvement, are able to help experts with their tedious and low-productive labeling tasks.

Atlas-based segmentation is a commonly used technique for segmenting medical images [25–28]. In atlas-based segmentation, it is non-trivial to apply non-linear image registration to a pair of images (i.e. a fixed target image and a moving atlas image). The transformed atlas image after registration, whose segmentation is usually given, propagates to the target image for anatomical labeling. Due to large variations in individual anatomical structures, it is hard to achieve perfect alignment between a pair of images.

Multi-atlas-based segmentation methods [22, 23, 29–36], which take advantage of having multiple atlases associated with manually delineated labels, have received growing interest in the past few years. The key assumption of the multi-atlas-based approach is that multiple atlases contain richer anatomical variability than a single atlas does. In multi-atlas-based segmentation, each atlas image is registered to align with a target image based on some similarity metrics; the annotated label map is transformed into the input image space; all the resulting transformed label maps are then combined to form the final segmentation for the input image, typically through label fusion mechanisms. Since multiple atlases incorporate inter-subject variability, more accurate and reliable segmentation results can be obtained through label fusion from multiple atlases. Many label fusion techniques have been widely investigated in the literature [31, 37–40]. Before applying any state-of-the-art label fusion method, it is non-trivial to introduce an atlas selection step to select the best candidate atlases that contribute to achieving better segmentation performance. The major drawback of multi-atlas segmentation is that it is computationally expensive. Therefore, the trade-off between accuracy and speed necessitates considerate algorithmic design in multi-atlas segmentation.

Deep learning methods have been adopted for multi-atlas [35, 41–44] and direct brain segmentation [5, 45–47]. However, most of the existing deep learning methods require a large amount of manually labeled data, substantial CPU training time and sophisticated fine-tuning in order to perform well on a dataset [46]. Furthermore, a neural network trained for image registration in an unsupervised learning manner is not easily comparable to traditional registration methods [48].

Inspired by the success of the 2D Scale Invariant Feature Transform (SIFT) flow approach in performing large-scale image matching [49–51] and 3D SIFT flow for atlas-based CT liver image segmentation [52], we propose a novel multi-atlas method for automated brain

structure segmentation in MRI volumes. As shown in Fig 1, the method consists of three sequential steps: registration, atlas selection and label fusion. In the registration stage, an integrated flow, based on grayscale and SIFT features, is introduced to generate voxel-wise correspondence between a pair of images. The SIFT descriptor [53] has been considered a milestone achievement in computer vision and has been widely adopted in a variety of applications such as object recognition [54], point tracking [55, 56], and panorama [57]. The SIFT descriptor is robust, to a certain degree, to local deformation, orientation, scaling, and illumination. SIFT has also been widely used in medical imaging [58–60]. At the same time, optical flow has already been successfully used in the medical imaging domain due to its efficiency and precision in mapping structures of interest [61]. Many models based on 2D as well as 3D optical flows have been developed and are available for medical image registration applications [62, 63], but the optical flow is only based on image intensity which sometimes fails to capture sufficient information between a pair of images. Apparently, the integrated flow, as an appropriate combination of optical flow and SIFT flow, is more robust to image variation and expected to produce better registration results. The integrated flow is incorporated into Markov random field (MRF) modeling and an improved sequential belief propagation optimization algorithm to generate a displacement flow field. In the second stage, the accompanying energy from the registration step is leveraged to select the best $K$ atlases for label fusion. Finally, a nonparametric label fusion approach [64] is adopted to extract information from flows, grayscale atlas images, SIFT images and annotations to generate the corresponding brain structure segmentation of the target image. The predicted segmentation can be used by pathologists for rapid positioning and further analysis of the subject's cerebral structures.

In our experiments, we apply our method to several well-known publicly available datasets, such as the Alzheimer's Disease Neuroimaging Initiative (ADNI) 1 Baseline 3T standardized dataset [65] and 2012 MICCAI Multi-Atlas Labeling Challenge brain image dataset [66], with several evaluation metrics such as the Dice coefficient and average Hausdorff distance. Choosing the patients' images as targets and the normal subjects' images as atlases in the ADNI dataset leads to a setting where there are large transformations and variations between images in the target dataset and that in the atlas set. The experiment on a composite dataset, in which the IBSR dataset [67] serves as targets and the EADC-ADNI dataset [68] serves as atlases, illustrates the effectiveness of our method in a cross-modality setting. We achieve comprehensive improvement in segmentation performance in comparison with widely used state-of-the-art methods such as Advanced Normalization Tools (ANTs) [69] symmetric image normalization registration (SyN) [70] method combined with Joint Label Fusion method [31, 71]. The registration time for a pair of images with our method is reduced to less than 1 minute in our processed datasets, which is estimated on a standard CPU node of the Microsoft High-Performance Computing (HPC) cluster. This is more than 7 times faster than the ANTs SyN registration method given the same input. Additionally, the execution time of our label fusion method grows linearly with regard to the number of participating atlases. The result of comparing different fusion methods shows that, given 135 candidate atlases, our label transfer method is 18 times faster than Joint Label Fusion [31] in the same pure CPU setting. It's noteworthy that, compared to the label fusion step in a multi-atlas segmentation system, registration is usually more time-consuming. The experimental results indicate that our method is sufficiently robust in dealing with large non-linear deformation, accurate in generating labels for target images, and efficient in terms of overall execution time.

A brief summary of our work is as follows:

1. A novel multi-atlas segmentation system based on the 3D integrated flow is proposed. The integrated flow is the essential element that connects all the components in the system.

2. We adopt an atlas selection method based on the final convergence value of the energy function in registration. This approach estimates the best candidate atlases for fusion, which costs almost no time because it only involves trivial sorting according to the numerical energy values from registration.

3. A coarse-to-fine flow matching approach and a 3D sequential belief propagation method with an improved message passing scheme are developed in registration. In addition, a simpler sequential belief propagation method is adopted in label fusion. These strategies not only make the system efficient but also lead to better segmentation results. Extensive experiments are conducted on five publicly available datasets under different settings. Our proposed method outperforms the competitive baseline methods in terms of efficiency and effectiveness.

## Method

In this section, we make a thorough description of our proposed system, which consists of the integrated 3D flow-based registration, the energy-based atlas selection and the integrated 3D flow-based label transfer.

### System overview

The core idea of our multi-atlas brain image segmentation system is inspired by the success of the 2D SIFT flow approach in performing large-scale image matching [49–51] and 3D SIFT flow for atlas-based CT liver image segmentation [52]. To segment a target image, we first register each image in the atlases to align with the target image based on the integrated flow. As a result, we obtain a dense correspondence and an accompanying energy value between each atlas image and the target with our integrated 3D flow-based registration method. Subsequently, we select the atlas images with the lowest matching energy as voting candidates, which are similar to the target images. With the candidate atlas images and their corresponding flows, we leverage the integrated 3D flow-based label transfer method to merge this information to generate a predicted segmentation volume for the target image. Nevertheless, many practical issues are to be resolved to build a reliable system for 3D MRI brain image segmentation.

As illustrated in Fig 1, the pipeline of our system consists of three main modules as follows:

- Registration: Establishing dense correspondence between the target image and each of the atlas images. Generating an integrated flow and an energy value for the correspondence of each pair of images. Warping the atlas images as well as their segmentations via the flows.

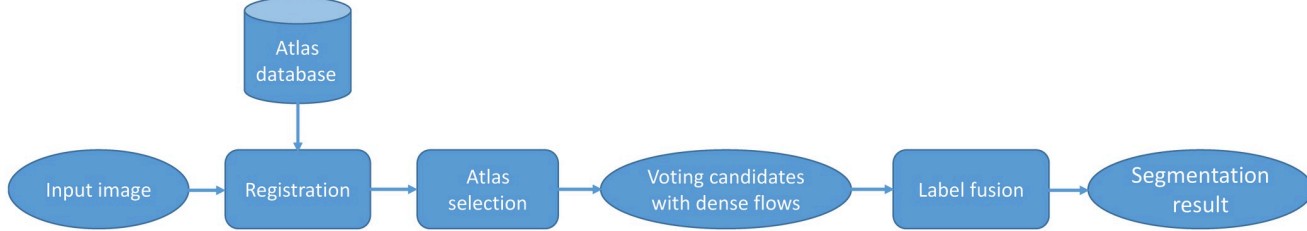

**Fig 1. A simple diagram for the pipeline of our proposed system.** The rectangles are three key components in our system: registration, atlas selection, and label fusion. Ovals and cylinders denote data representations.

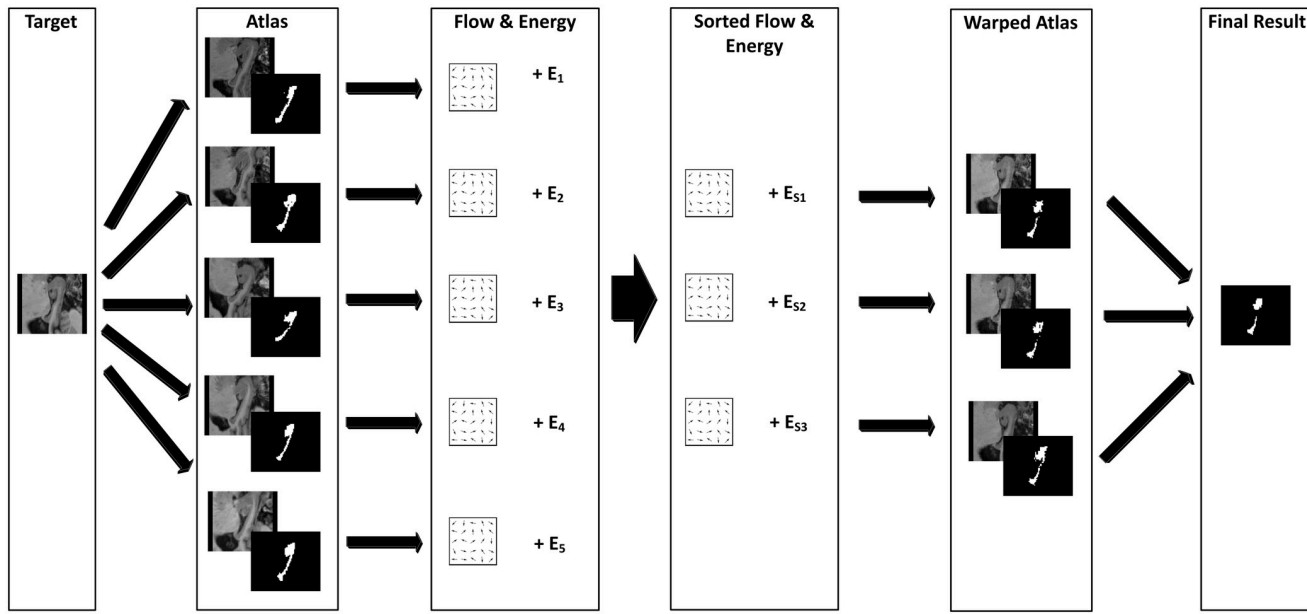

**Fig 2. Flowchart for our integrated 3D flow-based multi-atlas segmentation system.** Initially, every atlas image volume is registered with the target image, producing a flow field and a corresponding energy value. After sorting the registration results by their energy values from lowest to highest, the top *K* flow fields are chosen and applied to warp the corresponding atlas images and annotations to obtain *K* candidates. Upon performing fusion with the candidates, the predicted segmentation for the target image is produced. For convenience, we use selected slices of MRI images to denote 3D MRI volumes.

- Atlas selection: Sorting the warped atlases and segmentations in ascending order based on the corresponding energy values. Choosing the first *K* of them as fusion candidates.

- Label fusion: Merging the selected warped candidates into final segmentation for the target image by reconciling labels with features, and imposing spatial smoothness as well as other constraints via the label transfer.

The detailed flowchart for our complete system design is shown in Fig 2. The pseudo-code of our multi-atlas segmentation method is shown in Algorithm 1.

**Algorithm 1** Integrated 3D Flow-based Multi-atlas Segmentation Algorithm

```
1: Input: Atlas Images I = {I₁, ..., Iₙ}, Atlas Labels L = {L₁, ..., Lₙ},
Target Image I_target
2: Output: Predicted Segmentation L_target
   // Calculate SIFT features for all the images:
3: S_target = CalculateSIFT(I_target)
4: for i = 1 → n do
5:    Sᵢ = CalculateSIFT(Iᵢ)
6: end for
   // Combine grayscale and SIFT features into an integrated feature
vector:
7: M_target = CombineFeatures(S_target, I_target)
8: for i = 1 → n do
9:    Mᵢ = CombineFeatures(Sᵢ, Iᵢ)
10: end for
   // Calculate an integrated flow and an energy value for each pair:
11: for i = 1 → n do
12:    Fᵢ, Eᵢ = IntegratedFlow(M_target, Mᵢ)
13: end for
   // Sort the indices by obtained energies:
14: ID_original = (1, ..., n)
```

15: $\mathbf{ID}_{\text{sorted}}$ = SortIndexByEnergy($\mathbf{ID}_{\text{original}}$, $\{E_1, \ldots, E_n\}$)
16: $\mathbf{ID}_{\text{candidate}}$ = FirstKElements($\mathbf{ID}_{\text{sorted}}$, $K$)
 **// Denote $\mathbf{ID}_{\text{candidate}}$ = $\{id_1, \ldots, id_K\}$:**
 **// Use the label transfer to fuse the labels of the top $K$ candidates:**
17: $L_{\text{target}}$ = LabelTransfer($M_{\text{target}}, \{M_{id_{1\ldots K}}, L_{id_{1\ldots K}}, F_{id_{1\ldots K}}\}$)

### Integrated 3D flow-based registration method

In this subsection, we provide a detailed description of the formulation of our integrated 3D flow-based registration method by introducing the underlying integrated feature descriptor, the objective function, a 3D message passing optimization method and a 3D coarse-to-fine flow matching scheme.

**Integrated dense feature descriptor.** Two types of voxel-wise features are extracted and used consistently throughout our approach, more specifically, in the integrated 3D flow-based registration, the energy-based atlas selection and the integrated 3D flow-based label transfer. The grayscale feature for a voxel in an MRI image is a single value that contains merely intensity information, which is inherently located in an MRI image as a default image representation. Since the grayscale feature is common and widely used, we focus on the other feature used in our system.

SIFT is a type of feature descriptor that captures the gradient information of an image on a local scale. The typical SIFT feature extraction algorithm involves scale-space extrema detection, keypoint localization and keypoint feature extraction. In our approach, we develop a dense correspondence image registration method [72], which suggests that we can ignore the detection part of the original algorithm and focus on the feature extraction part. Similar to [52, 59, 60], we compute the gradient magnitude and orientation for each voxel $(x, y, z)$ as follows:

$$
\begin{aligned}
m(x, y, z) &= \sqrt{G_x^2(x, y, z) + G_y^2(x, y, z) + G_z^2(x, y, z)}, \\
\theta(x, y, z) &= \tan^{-1} \frac{G_y(x, y, z)}{G_x(x, y, z)}, \\
\phi(x, y, z) &= \tan^{-1} \frac{G_z(x, y, z)}{\sqrt{G_x^2(x, y, z) + G_y^2(x, y, z)}}, \\
G_x(x, y, z) &= G(x + 1, y, z) - G(x - 1, y, z), \\
G_y(x, y, z) &= G(x, y + 1, z) - G(x, y - 1, z), \\
G_z(x, y, z) &= G(x, y, z + 1) - G(x, y, z - 1),
\end{aligned}
\tag{1}
$$

where $G(x, y, z)$ denotes the intensity value in location $(x, y, z)$, $G_x$, $G_y$, and $G_z$ are gradients computed as finite difference approximations, whereas $m$, $\theta$, $\phi$ represent the magnitude and angular coordinates respectively. Intuitively, $(G_x, G_y, G_z)$ is a vector representing an approximate sub-gradient in $(x, y, z)$, and $(m, \theta, \phi)$ is its spherical coordinate, e.g., $m$ is the magnitude, $\theta$ is the azimuthal angle and $\phi$ is the polar angle.

With each voxel's magnitude and orientation obtained, we compute a histogram for each voxel. In this paper, we consider the neighborhood of a voxel to be an 8×8×8 cube, with the voxel being in the center, where we choose the one with the smallest coordinates by convention. We observe that the chosen neighborhood size is good enough in the experiments and the performance gain brought by increasing it does not outweigh the increased memory usage. The cube is further divided into eight $4 \times 4 \times 4$ sub-blocks. A sub-histogram for each sub-block is generated based on the magnitudes and orientations of the 64 voxels it contains. For simplicity and the best performance in our experiments, exactly 6 bins, denoting 6 directions,

**Table 1. Coefficients of the filter used in computing the histogram in a cell for SIFT features.**

| Offset X | Offset X | Offset Z | Coefficient |
|---|---|---|---|
| ±1 | ±1 | ±1 | $0.25^3$ |
| 0 | ±1 | ±1 | $0.25^2$ |
| ±1 | 0 | ±1 | $0.25^2$ |
| ±1 | ±1 | 0 | $0.25^2$ |
| 0 | 0 | ±1 | 0.25 |
| ±1 | 0 | 0 | 0.25 |
| 0 | ±1 | 0 | 0.25 |
| 0 | 0 | 0 | 1 |

where there are two opposite directions for each of the three dimensions, are adopted to fit the orientations into histograms. A gaussian weighting function is applied to the sub-histogram of each sub-block to account for their importance to the center voxel so that the farther one has less contribution to the weighted histogram, with coefficients shown in Table 1. The 6 histogram bins simply divide the space according to the angular coordinates $\theta$ and $\phi$, in which, for instance, $\theta \in \left[\frac{\pi}{4}, \frac{3\pi}{4}\right]$, $\phi \in \left[-\frac{\pi}{4}, \frac{\pi}{4}\right]$ together represent one of the bins. Such division is not disjoint (e.g., $\theta \in \left[\frac{\pi}{4}, \frac{3\pi}{4}\right]$, $\phi \in \left[-\frac{\pi}{4}, \frac{\pi}{4}\right]$ overlaps with $\theta \in [0, 2\pi]$, $\phi \in \left[\frac{\pi}{4}, \frac{3\pi}{4}\right]$) thus introducing overlap among bins because this is not the geographical way of creating parallels and meridians, which instead can perfectly create a partition of the space of all vectors based on their directions. However, our dividing strategy is intuitive and more computationally efficient. To overcome rotation dependence, we subtract the dominant orientation of the center voxel from all the orientations in the histogram so that the dominant one points to ($\theta = 0$, $\phi = 0$) and rotation invariance is thus guaranteed, which is equivalent to rotating an object to its most prominent gradient direction. In the end, the histograms of the eight sub-blocks are concatenated altogether, which ends up with a 48-dimensional SIFT feature vector for the center voxel. An illustration of one similar SIFT descriptor transformation with a $4 \times 4 \times 4$ neighborhood is shown in Fig 3.

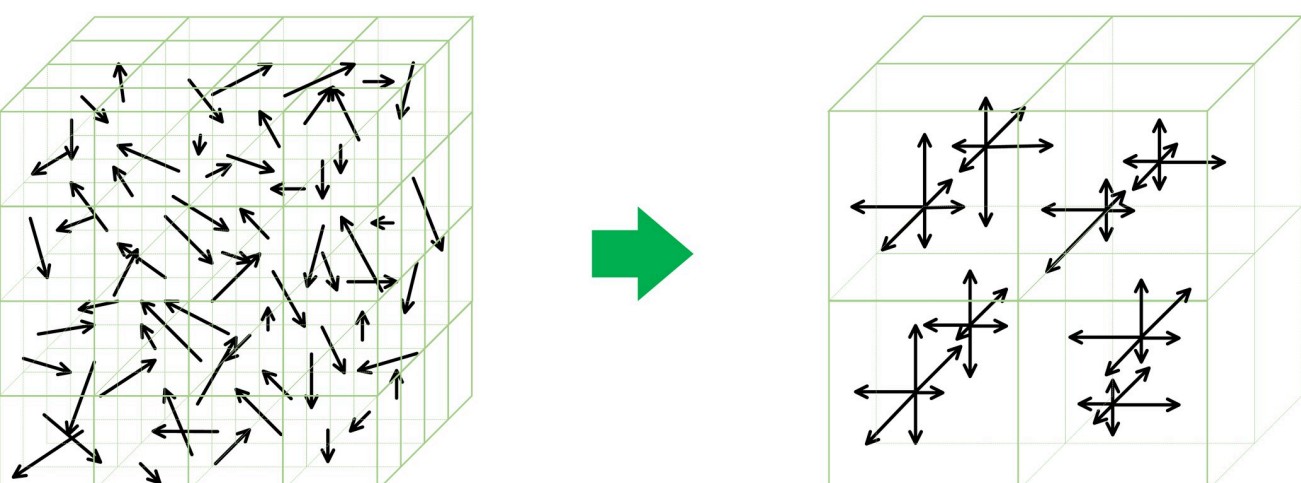

**Fig 3. Illustration of 3D SIFT descriptors.** The large arrow in the center points from a diagram showing the gradient distribution for a voxel's $4 \times 4 \times 4$ neighborhood to a diagram showing how the distribution fits into eight $2 \times 2 \times 2$ sub-blocks with 6 directions/histogram bins.

By experiments, we find that the SIFT features and intensities include complementary information for voxel-wise alignment. Hence, we adopt an integrated per-voxel feature vector consisting of the 48-dimensional SIFT descriptor and the grayscale value, which is formulated as

$$I(x, y, z) = S(x, y, z) \oplus \zeta G(x, y, z)$$
$$S(x, y, z) = (S_1(x, y, z), \ldots, S_{48}(x, y, z)),$$

(2)

where $I(x, y, z)$, $S(x, y, z)$, $\oplus$, $\zeta$ denote the integrated feature vector of dimension 49 for voxel at location $(x, y, z)$, the 48-dimensional SIFT feature vector, vector concatenation and a trade-off coefficient respectively. The introduction of $\zeta$ results from the fact that neither SIFT nor grayscale alone can capture the essential information for a voxel. For example, the SIFT feature is well suited to capturing image structures and voxel contexts, while grayscale values provide complementary raw intensity information in an MRI image when SIFT feature fails in alignment between two voxels because of gradient field homogeneity. The coefficient $\zeta$ also helps prevent inconsistency after feature vector normalization.

**3D integrated flow algorithm.** Based on the integrated dense feature descriptor explained above, in this work, we come up with an integrated 3D flow-based registration method. The motivation is that the 2D SIFT flow method performs well in alignment between complicated and spatially dissimilar scenes [49], as well as the success of the widely adopted optical flow [73] in image registration. Moreover, 3D SIFT flow with only SIFT features works well in 3D CT liver image segmentation [52]. A flow-based registration method is expected to produce better matching results with more than one single type of feature.

In our method, we have a mixed feature descriptor for each voxel, so our objective is to estimate the correspondence between a pair of images in which every voxel is replaced by a multi-dimensional feature vector. Let $\mathbf{p} = (x, y, z)$ represent the spatial coordinate of a voxel, and $\mathbf{f}(\mathbf{p}) = (u(\mathbf{p}), v(\mathbf{p}), w(\mathbf{p}))$ being the flow vector at $\mathbf{p}$. We denote by $I_1$ and $I_2$ the per-voxel integrated feature descriptors for two images, while $\varepsilon$ is a set of all the spatial neighborhood pairs (a six-neighbor system is used in this paper). To obtain the estimated registration displacement field, we adopt an MRF approach so that the objective is to minimize an energy function as follows:

$$
\begin{aligned}
E(\mathbf{f}, I_1, I_2) = \ & \sum_{\mathbf{p}} \min(\|I_1(\mathbf{p}) - I_2(\mathbf{p} + \mathbf{f}(\mathbf{p}))\|, t) + \\
& \sum_{\mathbf{p}} \eta(|u(\mathbf{p})| + |v(\mathbf{p})| + |w(\mathbf{p})|) + \\
& \sum_{(\mathbf{p}, \mathbf{q}) \in \varepsilon} [\min(\alpha|u(\mathbf{p}) - u(\mathbf{q})|, d) + \\
& \min(\alpha|v(\mathbf{p}) - v(\mathbf{q})|, d) + \\
& \min(\alpha|w(\mathbf{p}) - w(\mathbf{q})|, d)] + \\
& \log Z,
\end{aligned}
$$

(3)

which consists of a data term, a displacement term and a smoothness term, from left to right. $Z$ is the partition function for normalizing the potentials to be a legal probability distribution. So finding the optimal flow can be interpreted as finding the most probable configuration in this MRF, namely maximum a posteriori (MAP) inference in the literature on graphical models. The data term forces the corresponding voxel pairs to be matched as similar in their features as possible. The second term, the displacement term, restricts the flow vector to be small in its Manhattan distance to the origin. The smoothness term at the end of Eq (3) constrains the flow vectors of adjacent voxels to be close so that two adjacent voxels in the atlas image are

aligned with spatially close correspondences. $t$ and $d$ are the thresholds of the truncated norms. $\eta$ and $\alpha$ are the weights of the displacement term and the smoothness term, respectively. Note that minimization of the weighted sum of the displacement term and the smoothness term ensures that the flow vector does not deviate from the zero vector drastically.

In this energy function, truncated L1 norms are adopted in both the data term and smoothness term to combat outliers and preserve flow discontinuities respectively. Intuitively, if the L1 norm in the smoothness term is not truncated, the final flow field will be smooth locally and globally. In general, the registration/mapping field observes regularity in the global smoothness, while encountering local discontinuity at the boundary of the anatomical structures. On one hand, our proposed method consists of a regularization term that encourages the smoothness of the field; on the other hand, the SIFT features capture the important anatomical structures that observe robustness and informativeness to be able to model local discontinuity in the registration map. The truncated norms thus control the level of allowed local discontinuity while global smoothness is always imposed with $\alpha > 0$. We give an example of the displacement fields produced by our method and ANTs in Fig 4 with weak and strong local smoothness. The histograms are based on the smoothness terms for all pairs of adjacent voxels with $\alpha = 1$ and $d = \infty$. Note that our method produces less smooth fields than ANTs because of the discrete nature of integrated flows. However, smoother displacement fields can be obtained by choosing larger $\alpha$ and $d$ (the middle column in Fig 4). The displacement fields of our method are smooth overall since 90% of the gradients of displacements are under 2.0 (the left column in Fig 4).

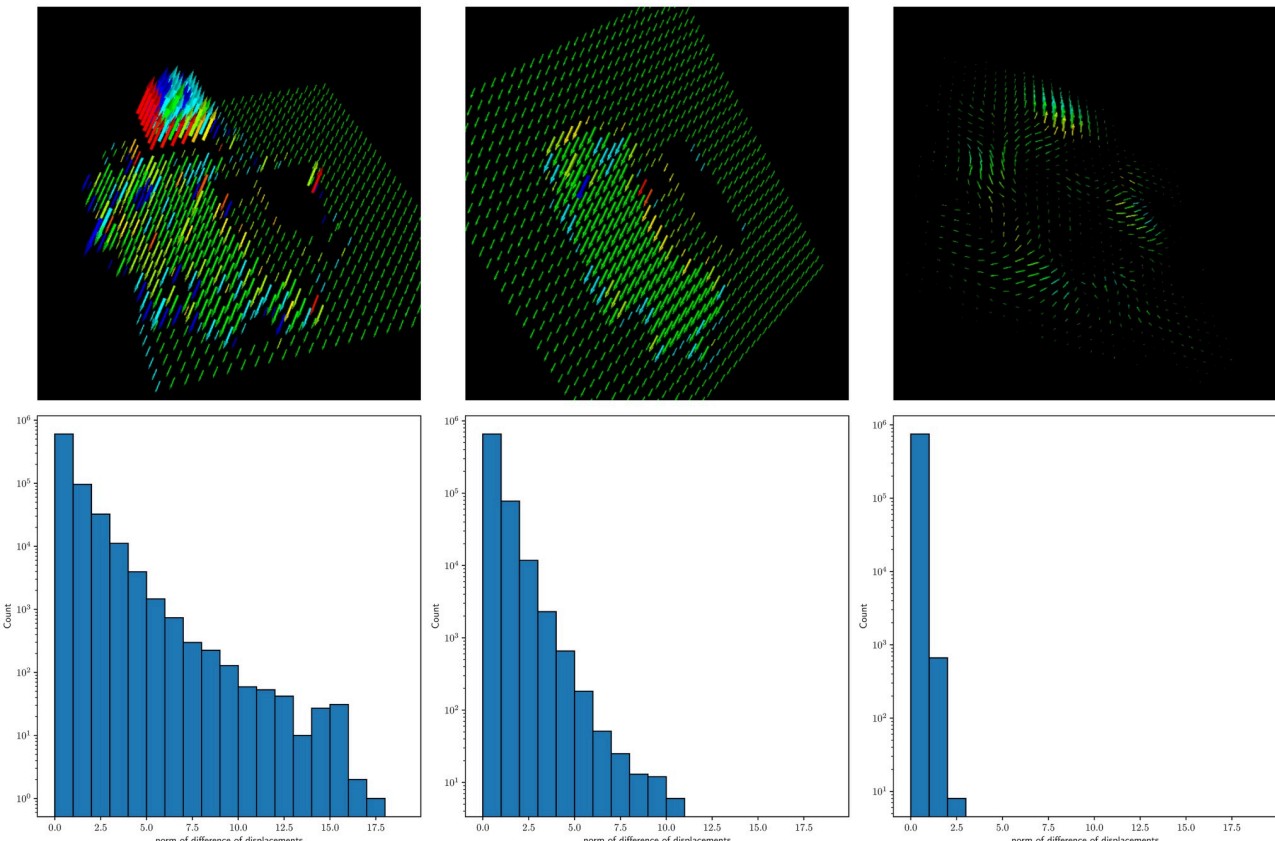

**Fig 4. Examples of the visualized displacement fields and their corresponding histogram statistics.** The histograms are based on the smoothness terms for all pairs of adjacent voxels with $\alpha = 1$ and $d = \infty$. From left to right: the integrated flow with less restriction on local smoothness; the integrated flow with stronger local smoothness; ANTs SyN.

**3D ternary-layer message passing for belief propagation.** Since classic MRF energy min-imization algorithms such as graph cuts [74] and max-product belief propagation [75] find local minima that are good approximations of global minima but are not efficient in practice, especially for large 3D image volumes in our case, we adopt a sequential belief propagation (BP-S) approach [49, 76–78] to optimize the above energy function. To avoid passing messages inefficiently in a six-neighbor system while still pursuing a good approximation for energy minimization, an appropriate message update schedule is required by exploring the special structure of a 3D image grid. As suggested by Liu et al. [49] and Shekhovtsov et al. [79], the smoothness term in the energy function in Eq (3) is decoupled in L1 norm form for three dimensions, which makes the MRF model decomposable in belief propagation. Hence, the message of a flow vector about the smoothness penalty can be divided into three individual parts for its three-dimensional components. Each part is passed around in an isomorphic 3D volume for its own dimension. In this way, the original volume becomes three isomorphic vol-umes. The messages of the smoothness term, which are called intra-layer messages, are propa-gated independently within the respective volumes, while the messages of the data term and displacement term that connect the corresponding duplicate nodes belonging to the three vol-umes, are called inter-layer messages, or ternary-layer messages. We use the term, layer, for the isomorphic volume when referring to inter-layer and intra-layer messages, which follows from the 2D dual-layer expression. With this setup and the help of Shekhovtsov et al. [79], an effi-cient 3D message passing scheme is designed as follows. During propagation, at the beginning of one iteration, we update inter-layer messages by passing them from the two counterpart vol-umes to the current volume. Afterwards, the intra-layer messages within the current volume are updated accordingly in which a forward updating routine is followed by a backward one in reverse order. This approach not only reduces time complexity and space complexity of the algorithm, but also makes registration results more robust and less sensitive to noise.

Formally, following from the notations in [79, 80], we define the graph as $G = (\mathcal{V}, \mathcal{E})$ and the node set as $\mathcal{V} = \mathcal{V}^X \cup \mathcal{V}^Y \cup \mathcal{V}^Z$, where $\mathcal{V}^X \sim \mathcal{V}^Y \sim \mathcal{V}^Z \sim \mathcal{V}^I$ for layer $X$, $Y$, $Z$ in three dimensions, with $\sim$ denoting isomorphism and $\mathcal{V}^I$ denoting the node set of the original image volume. The edge set is $\mathcal{E} = \mathcal{E}^X \cup \mathcal{E}^Y \cup \mathcal{E}^Z \cup \mathcal{E}^{XY} \cup \mathcal{E}^{XZ} \cup \mathcal{E}^{YZ}$, in which $\mathcal{E}^X$ is a set of adjacent voxel pairs for $\mathcal{V}^X$ in a six-neighbor system and $\mathcal{E}^{XY} = \{(p^X, p^Y)|p^X \in \mathcal{V}^X, p^Y \in \mathcal{V}^Y, p^X \sim p^Y\}$ represents the inter-layer edge set for any pair of isomorphic nodes that shares the same loca-tion $p$ in the original volume but in different layers. In addition, $\mathcal{E}^X(p) = \{q \in \mathcal{V}^X|(p, q) \in \mathcal{E}^X\}$ stands for the set of neighbors of node $p$ in $\mathcal{V}^X$, where $(p, q)$ is treated as an unordered pair. The superscript of $p^X$ is omitted for conciseness when the context is clear. Let $m_{pq}^X$ be the intra-layer message passed from node $p$ to $q$ in $\mathcal{V}^X$, $m_p^X$ be the inter-layer message for node $p$ in $\mathcal{V}^X$, $x_p$ be the label assigned to $p$, in which the label encodes the offset in registration, and $x_p^{X*}$ be the optimal solution for $p$ in layer $X$. Generally, the message update process and the belief vector as well as the optimal assignment can be formulated as

$$
\begin{aligned}
m_p^X(x_p) &= \min_{y_p, z_p} \left( D_p(x_p) + \sum_{q \in \mathcal{E}^Y(p)} m_{qp}^Y(y_p) + \sum_{q \in \mathcal{E}^Z(p)} m_{qp}^Z(z_p) \right) \\
m_{pq}^X(x_q) &= \min_{x_p} \left( V(x_p, x_q) + m_p^X(x_p) + \sum_{t \in \mathcal{E}^X(p)\setminus q} m_{tp}^X(x_p) \right) \\
b_p^X(x_p) &= V(0, x_p) + m_p^X(x_p) + \sum_{q \in \mathcal{E}^X(p)} m_{qp}^X(x_p) \\
x_p^{X*} &= \min_{x_p} b_p^X(x_p),
\end{aligned}
\tag{4}
$$

where $V(x_p, x_q)$ denotes the smoothness penalty or discontinuity cost and $V(0, x_p)$ accounts for displacement cost in belief computation, with unary term $D_p(x_p)$ being the data cost. In this framework, we leverage the min-sum belief propagation [80] and compute the minimum marginal whenever the messages meet at a node. Message passing in these isomorphic volumes is performed in the order of $X$, $Y$, $Z$. Because of the truncated L1 norms in the energy function, distance transform techniques in [80] is adopted during intra-layer message computation. All the messages are initialized to zero before the first iteration. In the first iteration, for instance, we update the inter-layer message for layer $X$ based on information from layer $Y$ and $Z$. After that, a forward passing starts from the node in $\mathcal{V}^X$ with the smallest indices, say, $(0, 0, 0)$, and updates the messages to the neighbor nodes whose indices are coordinate-wise larger than this node, for example, $(0, 0, 1)$ and $(1, 1, 1)$. The forward updating process scans the nodes in $\mathcal{V}^X$ in lexicographical order of the indices while the backward updating that follows operates on the nodes in a totally reverse order. The subsequent iterations update messages for layer $Y$ and $Z$ similarly and go back to layer $X$. Normalization is applied to the computed messages for better convergence. The final optimal solution is simply the label assignment with the smallest belief value, or equivalently, the maximum a posteriori (MAP) estimate.

**3D coarse-to-fine flow matching.** Generally speaking, the integrated 3D flow-based registration method matches a voxel in the fixed image with any voxel in the moving image, which indicates at least $O(N^6)$ time complexity in terms of a 3D image's dimensional length $N$. Hence it is not feasible in practice, especially in 3D medical image analysis, typically with $N > 100$.

Motivated by [49, 81], to speed up the matching and registration process, we develop a 3D coarse-to-fine window searching approach [82] for our registration method. The main idea is similar to the classic divide-and-conquer approach but in an approximate probabilistic algorithm setting. Initially, the original image is downsampled several times, which leads to a hierarchical structure of images from the finest to the most compressed. At each level in the hierarchy, an individual matching window is set up for every voxel based on the resulting flow vector from last level. This coarse-to-fine flow matching scheme computes the flow at a coarse level of an image volume, while gradually propagating and refining the flow from coarse to fine. More formally, assume that the fixed image $I$ is to be downsampled $h$ times, resulting in $h + 1$ images, $I^{(0)}, I^{(1)}, \ldots, I^{(h)}$, in which $I^{(0)}$ is the original fixed image $I$ and for any $k > 0$, $I^{(k)}$ is a downsampled image of $I^{(k-1)}$. Consider a fixed image's voxel $\mathbf{p}^{(h)} = (x, y, z)$ to be matched in the coarsest level and the current flow vector for it is initialized to be $(0, 0, 0)$. According to the energy function, the best matching flow vector $\mathbf{f}^{(h)}(\mathbf{p}^{(h)}) = (u^{(h)}(\mathbf{p}^{(h)}), v^{(h)}(\mathbf{p}^{(h)}), w^{(h)}(\mathbf{p}^{(h)}))$ is found within a searching window centered at $\mathbf{c}^{(h)}(\mathbf{p}^{(h)}) = \mathbf{p}^{(h)} + (0, 0, 0)$, whose size is fixed to be $W \times W \times W$ in our method. The voxel and centroid of the searching window are all updated and upsampled to their corresponding new coordinates during propagation to the finer level. The coarse-to-fine propagation process can be formulated as

$$
\begin{aligned}
I^{(0)} &= I \\
I^{(k)}(x, y, z) &= \sum_{dx, dy, dz \in \{0,1\}} I^{(k-1)}(2x + dx, 2y + dy, 2z + dz)/8 \\
\mathbf{c}^{(h)}(\mathbf{p}^{(h)}) &= \mathbf{p}^{(h)} \\
\mathbf{c}^{(k-1)}(\mathbf{p}^{(k-1)}) &= 2(\mathbf{p}^{(k)} + \mathbf{f}^{(k)}(\mathbf{p}^{(k)})) \quad \forall k \in [1, h],
\end{aligned}
\tag{5}
$$

where $\mathbf{p}^{(k-1)}$ denotes the corresponding upsampled voxels of $\mathbf{p}^{(k)}$. So the flow vector is refined from a coarse level to the original volume while updated in a searching window at each level. It is noteworthy that the window size $W$ should be at least the dimensional length of the coarsest image for global voxel matching, but should not be too large for stable searching in fine levels.

Downsampling without pre-filtering may cause aliasing. However, in our case, the coarse-level downsampled image simply serves as a small-sized image with aggregated intensity information of the original image in order to compute a good starting point for the fine-level image to search for more accurate correspondences. Moreover, we smooth the volumes prior to message passing and generating coarser levels in belief propagation.

## Energy-based atlas selection

Since the population represented by the atlases is typically heterogeneous, in terms of age, gender and morphology, it would be better to only propagate and combine certain atlases analogous to the target image instead of the whole atlas database. In our study, a flow vector and the corresponding matching energy between a pair of images are obtained by optimizing the energy function in the registration step. Based on observation and analysis, low matching energy indicates a higher similarity between the target image and the atlas image. Therefore, in order to achieve a higher accuracy in final segmentation results, we develop an effective atlas selection strategy based on the registration energy. The basic procedures of our atlas selection strategy can be summarized as follows: (i) the set of flows and atlases for the target image are sorted in increasing order based on their corresponding registration matching energy. (ii) the top $K$ flows and their corresponding original atlas images are selected from the sorted sequence. The optimal value for the number of candidates $K^*$ can be determined by cross-validation or a judicious choice.

There are two advantages of our energy-based atlas selection strategy. First, unlike the sophisticated fine-grained Support Vector Machine (SVM) Rank [83], our method leverages an optimization energy value as the similarity measure, which is usually the final objective value of an optimization method, and efficiently filters out those potentially unbeneficial atlases with large energy values, which is almost cost-free in terms of time. Second, although our atlas label fusion method is linear in the number of candidate atlases as discussed in the experiments, some label fusion methods such as Joint Label Fusion [31] explore the inter-dependency among candidate atlases and involve computation of the inverse of a large matrix, which leads to a quadratic time complexity with a non-trivial constant that is not negligible in practice. Since the practical execution time of such label fusion methods grows much faster than that of a linear method, which is studied in the experiments, it is of great significance to deal with only a few candidate atlases by means of our atlas selection method instead of computing a label fusion result for one hundred atlases.

## Integrated 3D flow-based label transfer method

With the selected candidate atlases and their flows, we can merge this information to predict a segmentation for the target image by fusion methods. In this paper, we develop an integrated 3D flow-based label transfer method inspired by the non-parametric scene parsing in natural images [64] and the 3D SIFT flow-based registration method [52]. As the name of the method indicates, it transfers all the warped labels of the atlas images into a single label image. At first, we have a target image $I$ and some atlases $\{I_1, I_2, \ldots, I_K\}$ with known annotations to help prediction, all of which are in the form of a multi-dimensional integrated feature vector. The complete candidate set is denoted by $\{I_i, L_i, \mathbf{f}_i\}_{i=1:K}$, where $I_i$ is one of the aforementioned atlases, $L_i$ is the corresponding annotation or segmentation image, and $\mathbf{f}_i$ is the flow vector from registration. After applying the flow vectors, a set of warped candidates, $\{I_i', L_i'\}_{i=1:K}$, is obtained. We aim to use the above information to generate $L$, the corresponding annotation for $I$, which is the final predicted segmentation for the target image. Similar to the previous registration method, we use a method based on the MRF model to estimate the final result. The objective

energy function takes into consideration the object's prior information, atlas data information and spatial smoothness information:

$$
\begin{aligned}
F(I, L, S') = \quad & \sum_{\mathbf{p}} \psi(L(\mathbf{p}); I, \{I'_i\}) + \\
& \alpha \sum_{\mathbf{p}} \lambda(L(\mathbf{p}); \{I'_i\}) + \\
& \beta \sum_{(\mathbf{p}, \mathbf{q}) \in \varepsilon} \phi(L(\mathbf{p}), L(\mathbf{q}); I) + \\
& \log Z,
\end{aligned}
\tag{6}
$$

where $Z$ serves the purpose of normalization and $\alpha$, $\beta$ are introduced to account for a trade-off among the three terms.

The first summation is the likelihood term, which is defined as follows:

$$
\psi(L(\mathbf{p}) = l; I, \{I'_i\}) =
\begin{cases}
\min_{i \in \Omega_{\mathbf{p},l}} \|I(\mathbf{p}) - I'_i(\mathbf{p} + \mathbf{f}(\mathbf{p}))\| & \text{if } \Omega_{\mathbf{p},l} \neq \emptyset, \\
\tau & \text{if } \Omega_{\mathbf{p},l} = \emptyset,
\end{cases}
\tag{7}
$$

where $\Omega_{\mathbf{p},l}$ is the index subset of the participating atlases in which the registered label is $l$ at voxel location $\mathbf{p}$. Note that if there is no atlas found to have label $l$ at the specific location, a threshold $\tau$ is assigned.

The prior term, after the likelihood term, simply accounts for the prior information of all candidate labels. We count the number of occurrences for each label at each location:

$$
\lambda(L(\mathbf{p}) = l; \{I'_i\}) = -\frac{\log \dfrac{\text{hist}_l(\mathbf{p})}{\max_{\mathbf{q}} \text{hist}_l(\mathbf{q})}}{\log \dfrac{\epsilon_{\text{prior}}}{K + \epsilon_{\text{prior}}}},
\tag{8}
$$

in which $\text{hist}_l(\mathbf{p})$ is the histogram of the occurrence of label $l$ at location $\mathbf{p}$ and $\epsilon_l$ is a parameter for prior information for label $l$.

The smoothness term is expanded into:

$$
\phi(L(\mathbf{p}), L(\mathbf{q}); I) = \delta(L(\mathbf{p}), L(\mathbf{q})) \left( \frac{\epsilon + e^{-(2\langle \|I(\mathbf{p}) - I(\mathbf{q})\|^2 \rangle)^{-1} \|I(\mathbf{p}) - I(\mathbf{q})\|^2}}{\epsilon + 1} \right),
\tag{9}
$$

where $\delta$ is a penalty weight function of the difference between two voxel labels, $\langle \cdot \rangle$ denotes an average operation over the whole image, $\| \cdot \|$ represents the L2 norm of a vector and $\epsilon$ is a parameter for scale modification.

Similar to the previous registration method, we adopt a simpler sequential belief propagation algorithm to optimize the energy function for faster convergence, in which a coarse-to-fine propagation algorithm is adopted as well.

## Asymptotic analysis

For complexity analysis, we assume that there are $N \times N \times N$ voxels in each image, $D$-dimensional image data ($D = 3$ in our case), $L$ types of labels, $E$ voxels in the defined neighborhood system, $F$ dimensions for a voxel's feature vector, $W$ voxels in the coarse-to-fine searching window, $M$ atlas images and $T$ iterations of message passing in belief propagation.

In the integrated 3D flow-based registration, the coarse-to-fine searching and the belief propagation process are the main contributors to the total execution time. Because of the divide-and-conquer characteristics of downsampling, according to master theorem [84], the recurrence relation gives us $O(DEWN^3F + TW^3N^3)$, where the first term accounts for message allocation preprocessing and the second term indicates the complexity of message passing and the overall coarse-to-fine scheme.

In atlas selection, common sorting algorithms would require a quasilinear complexity $O(M \log M)$.

As for the label transfer method, the preprocessing of the candidate set takes $O(MLFN^3)$ while the complexity of the message passing is $O(TDELN^3)$. Thus the time complexity of the label transfer is $O(MLFN^3 + TDELN^3)$.

Above all, the overall time complexity to compute predicted segmentation for one target image with our proposed method is $O(N^3(DEWF + TW^3 + MLF + TDEL))$, which is linear with respect to the atlas size and number of labels. In practice, however, an image is usually 2D or 3D, and the neighborhood size is usually 4 or 6, which makes $D$ and $E$ relatively small constants. In addition, $T < 100$ and $F = 49$ could be considered small constants as well compared to the searching window size and image volume dimensional length. By this simplification, the complexity can be written as $O(N^3L(W^3 + M))$. This form of complexity lacks some details but clearly points out the main contributors when the image size and atlas set size outweigh other factors. Thus, our method is expected to be scalable and applicable in large-scale datasets, which is further empirically validated in our experiments.

## Results

In this section, we elaborate on the datasets, preprocessing steps, implementation details, and the evaluation metrics adopted in our experiments. Subsequently, we report the results of comparing different systems and study the influence of individual components.

### Datasets and preprocessing

Our proposed method is thoroughly evaluated on five publicly available brain image datasets. A summary of the relevant information to our experiments on these datasets is shown in Table 2 and a detailed description of them is given as follows.

1. **ADNI**: Data used in the preparation of this article were obtained from the Alzheimer's Disease Neuroimaging Initiative (ADNI) database (adni.loni.usc.edu). The ADNI was launched in 2003 as a public-private partnership, led by Principal Investigator Michael W.

**Table 2. Summary of the datasets in our experiments.**

| Dataset | No. of subjects | Target set | Atlas set | Age | Diagnosis | Chosen annotation |
|---------|----------------|------------|-----------|-----|-----------|-------------------|
| ADNI | 151 | 33(AD)+71(MCI) | 47(NC) | 55-90 | AD/MCI/NC | Hippocampus |
| MICCAI 2012 | 35 | 20 | 15 | 34.16 ± 20.40 | AD/MCI/NC | Anterior cingulate gyrus |
| LPBA40 | 40 | LOOCV | 40 | 29.20 ± 6.40 | Normal | Cuneus |
| Hammers | 30 | LOOCV | 30 | - | Normal | Amygdala |
| IBSR-EADC | 18+135 | 18 | 135 | 7-71, 55-90 | AD/LMCI/MCI/NC | Hippocampus |

Shown dataset characteristics include the number of subjects in the dataset, the number of volumes in our split target set and atlas set, age, diagnosis, and the chosen structure segmentation for experiments. Since IBSR-EADC contains two datasets, the age information is separated by a comma. LOOCV: leave-one-out cross-validation. AD: Alzheimer's disease subjects. MCI: mild cognitive impairment subjects. NC: normal control healthy subjects. LMCI: late mild cognitive impairment subjects.

Weiner, MD. The primary goal of ADNI has been to test whether serial magnetic resonance imaging (MRI), positron emission tomography (PET), other biological markers, and clinical and neuropsychological assessment can be combined to measure the progression of mild cognitive impairment (MCI) and early Alzheimer's disease (AD). For up-to-date information, see www.adni-info.org.

In this paper, we use the ADNI1:Baseline 3T dataset in the ADNI 1 Standardized Data Collections [65]. The dataset consists of MRI scans acquired from 151 subjects, aged 55 to 90, who received 3.0-T scans and passed quality checks under the control of the ADNI MRI Core. The resulting dataset has one screening scan for each subject. More specifically, there are a total of 151 T1-weighted 3.0-T MRI scans from 33 Alzheimer's Disease subjects (AD), 71 Mild Cognitive Impairment subjects (MCI) and 47 Normal Control healthy subjects (NC). Manual whole-brain segmentations are provided by experts. The initial size of the image volumes is $256 \times 256 \times 256$ voxels with a voxel resolution of $1.0 \times 1.0 \times 1.0$ mm$^3$. All the scans in the dataset are used in our experiments. We split the dataset so that the 47 NC images form the atlases (i.e. the training set), while the 33 AD and 71 MCI images form the target set (i.e. the test set). The splitting strategy creates a challenging multi-atlas segmentation dataset in which large deformation is required in registration. We choose the hippocampus as the structure to be segmented in this dataset.

2. **MICCAI 2012**: The MICCAI 2012 Multi-Atlas Labeling Challenging brain image dataset [66] makes up of 35 T1-weighted brain MRI scans obtained from the Open Access Series of Imaging Studies (OASIS) project [85]. The dataset comes with segmentation provided by Neuromorphometrics, Inc. (http://Neuromorphometrics.com/) using the brain COLOR labeling protocol. All the MRI scans are of size about $256 \times 256 \times 300$ voxels and resolution $1.0 \times 1.0 \times 1.0$ mm$^3$. We follow the same settings of the challenge in which the test set includes 20 images and the training set includes 15 images. Anterior cingulate gyrus is the chosen segmentation structure for the MICCAI 2012 dataset.

3. **LPBA40**: The LPBA40 dataset [86] provided by the Laboratory of Neuro Imaging (LONI) consists of 40 T1-weighted brain MRI scans of normal subjects. The subject group consists of 20 males and females, of age $29.2 \pm 6.4$ years. 124 contiguous coronal slices are acquired on a GE 1.5T system with 1.5 mm thickness and in-plane voxel resolution of $0.78 \times 0.78$ mm$^2$ for 2 subjects or $0.86 \times 0.86$ mm$^2$ for 38 subjects. More than fifty structure delineations are available, including 50 cortical structures, 4 subcortical areas, the brainstem and the cerebellum, among which we choose the cuneus as the segmentation target. Since LPBA40 is a brain MRI image dataset of healthy people, we perform a leave-one-out cross-validation on it, in which one volume is chosen to be the target image with the remaining 39 volumes being the atlas set in each round.

4. **Hammers**: The Hammers adult brain atlases [87, 88] are generated by manual tracing of 83 anatomical structures based on MRI scans from 30 healthy adult subjects. Several registration and statistical analysis steps are performed to produce individual atlases, a probabilistic atlas and a maximum probability map. We adopt leave-one-out cross-validation similar to LPBA40 and choose the amygdala as the target structure for segmentation.

5. **IBSR-EADC**: The Internet Brain Segmentation Repository (IBSR) dataset [67] consists of 18 T1-weighted MRI volumetric images, provided by the Center for Morphometric Analysis at Massachusetts General Hospital. The subjects are 14 males and 4 females, whose ages range from 7 to 71 years old. In this version of the IBSR dataset, the slices are 1.5 mm apart with an in-plane resolution of 0.8371 mm, 0.9375 mm, or 1.000 mm. More than 100 cortical and subcortical parcellations are available in this dataset.

European Alzheimer's Disease Consortium (EADC) and ADNI have made an effort to provide a consensual, harmonized protocol (HarP) [89–91] for MRI scan hippocampus segmentation. Labels of 135 ADNI images were released by Boccardi et al. in 2015 [68]. T1-weighted structural MR images from 135 subjects are acquired by MP-RAGE with a thickness of 1.2 mm and the acquisition plane as sagittal. Both 1.5T and 3T scans from subjects aged 55-90 years are included in the dataset. Among the 135 subjects, 45 of them are AD subjects, 17 are late MCI (LMCI) subjects, 29 are MCI subjects and the remaining 44 are NC subjects, or normal people.

We create a composite dataset, called IBSR-EADC, of the above two datasets, in which the 18 images of the IBSR dataset form the test set and the 135 images of the EADC ADNI HarP dataset are defined as the training set. Since hippocampal volume is the only available annotation in the EADC ADNI HarP dataset training labels, we perform hippocampus segmentation on the IBSR-EADC dataset.

The ADNI images in the standardized dataset are of good quality and well preprocessed, including intensity normalization and inhomogeneity correction, so it is not indispensable for us to perform more preprocessing operations like denoising and normalization before registration. For the other datasets, MICCAI 2012, LPBA40, Hammers and IBSR-EADC, we apply N4 Bias Field Correction [92] to all the whole brain MRI volumes.

To make better comparisons and reduce the image size, $64 \times 64 \times 64$ sub-volumes with the same resolution as the original volume and a single annotated brain structure in the center, are cropped from the original volumes. The bounding box surrounding each structure is obtained according to the region of interest (ROI) generated by Freesurfer [28]. The guidance of Freesurfer helps lower memory consumption and produce more accurate results. As shown in Fig 5, each image volume produces two relatively small volumes. Consequently, each original dataset is split into two disjoint datasets with structures on the left and right evaluated independently. For instance, in the ADNI dataset, 33 AD, 71 MCI and 47 NC left hippocampus sub-volumes form a complete target and atlas dataset, while the 33 AD, 71 MCI and 47 NC right hippocampus sub-volumes constitute another target and atlas dataset. The final segmentation results are averaged over the whole dataset including the left structure dataset and the right structure dataset.

## Implementation details

Our proposed methods and baseline methods are evaluated on the same datasets. During each experiment on a dataset, we take the target images from the test set one by one, take all the images in the training set as atlases, apply a multi-atlas segmentation method to obtain segmentation for the target image and evaluate the segmentation. All the images in the test set are processed and evaluated independently.

For visualization of the 3D MRI volumes, we use the ITK-SNAP software [93] and the Free-Surfer suite [28].

We implement the proposed integrated 3D flow-based multi-atlas segmentation method in pure C++ without any external library. An ANTs version 2.1.0 Windows release package [69] as well as its auxiliary tools such as SyN registration [70], Joint Label Fusion [31] and N4 Bias Field Correction [92], is used in our experiments. As for Elastix [94, 95], a 64-bit Windows release version 4.8 package is employed. The Learning to Rank method [96] is reimplemented based on the code provided by the author of the method, with ANTs SyN as the registration method and Joint Label Fusion as the fusion method. The STAPLE fusion algorithm [97] executable comes from Computational Radiology Kit (CRKIT). The atlas selection algorithm after registration is implemented in C#. The final values of our energy function in registration serve

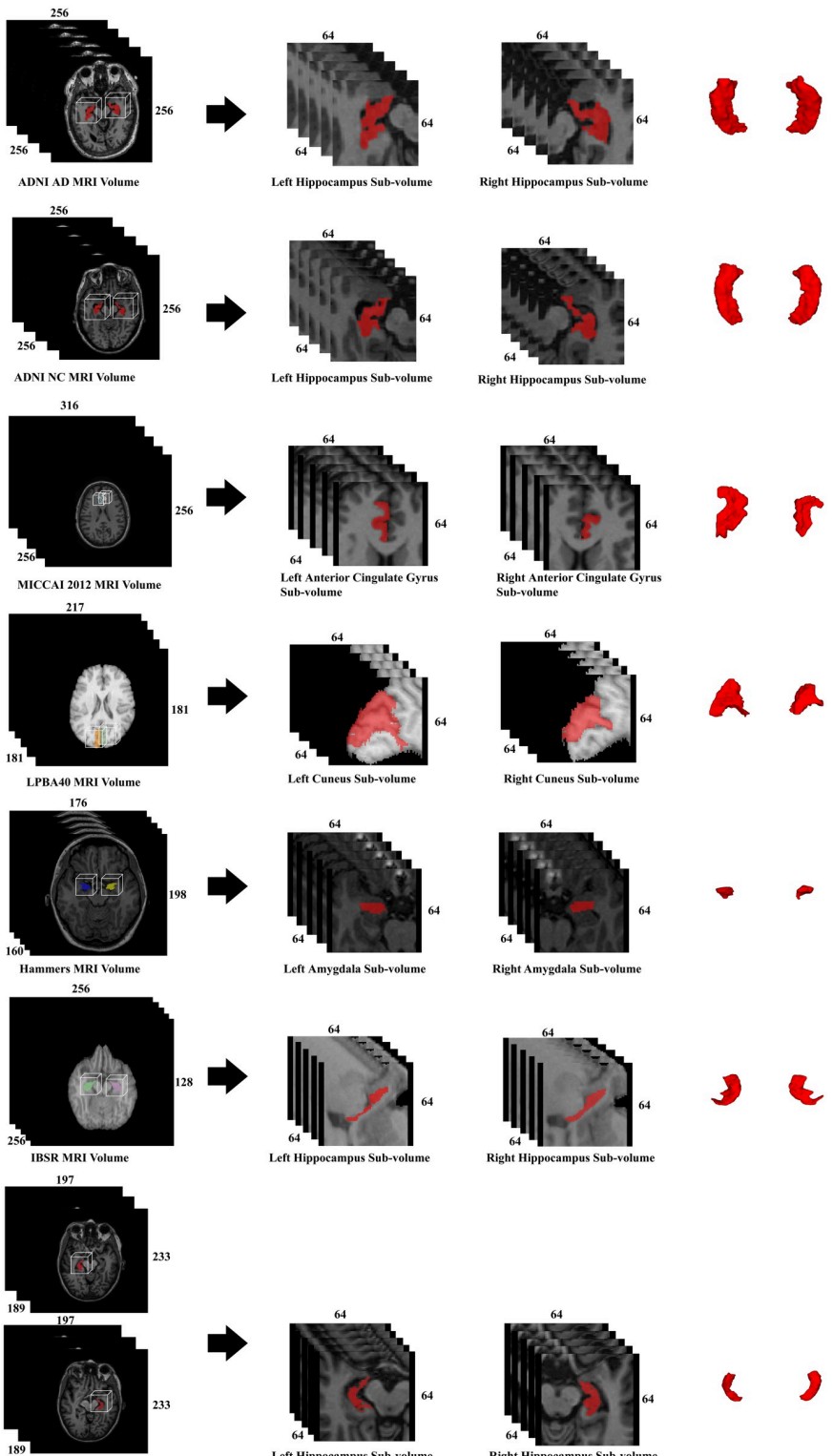

**Fig 5. Volume examples of the datasets in our experiments.** For each MRI volume, we extract the left and right sub-volumes of some brain structure to obtain a dataset of cropped volumes according to ROIs generated by Freesurfer. The dimensional length is annotated alongside the volumes shown in stacked slices.

as the atlas selection criteria for our method. The final metric values given in ANTs SyN and Elastix after convergence or termination are adopted in their atlas selection process, respectively. Evaluation of segmentation results is done with the EvaluateSegmentation Tool [98] provided by Visual Concept Extraction Challenge in Radiology (VISCERAL). Above all, we write shell and Python scripts for launching all the programs at a high level. All the experiments are conducted on a Microsoft HPC cluster with 2 Quad-core Xeon 2.43 GHz CPUs for each compute node.

All the hyper-parameters of our proposed methods are estimated by cross-validation on the ADNI training set and reused in all the following experiments. In the integrated 3D flow-based registration, $t = \infty$ (no restriction on data terms with good preprocessing), $\eta = 0.005$, $\alpha = 2$, $d = 40$, the number of iterations in BP-S is 60, the searching window size is $5 \times 5 \times 5$ and the height of hierarchy in 3D coarse-to-fine flow matching is set to 3, with $\zeta = 2$ for the integrated dense feature vector. The number of candidate atlases in atlas selection is set to $K = 15$ if there are more than $K = 30$ atlases, and set to half of the atlas set size otherwise. In the integrated 3D flow-based label transfer, $\alpha = 5$, $\beta = 0.9$, $\epsilon = 0.3$, $\epsilon_{prior} = 0.2$ and $\tau = 500$. For ANTs and Joint Label Fusion, we take advantage of the hyper-parameters in [99]. The default parameter settings are adopted in Elastix, Learning to Rank and any other unmentioned methods.

## Evaluation methods

Three evaluation metrics for volumetric segmentation are adopted in this paper, including the Dice coefficient, average Hausdorff distance and Cohen's kappa coefficient.

The Dice coefficient (DC) [100], also known as the Dice similarity coefficient (DSC), the Sørensen index or the F1 score, is one of the most widely used metrics for 3D medical image segmentation. DC measures the reproducibility or the spatial overlapping ratio of two segmentation volumes. It is defined as:

$$\mathrm{DC}(G, S) = \frac{2|G \cap S|}{|G| + |S|} \times 100\%, \tag{10}$$

where $G$ is the set of spatial voxel positions in the ground truth segmentation and $S$ is the segmentation result or prediction. $\cap$ denotes intersection and $|G|$ represents the cardinality of $G$. A higher DC value implies a more accurate segmentation result.

The Hausdorff distance (HD) measures the spatial distance between two sets of points, or two segmentation volumes in the context. Because of its own characteristics, HD is asymmetric, directed and sensitive to outliers. Thus, we adopt the average Hausdorff distance (AVD) [98], which is less sensitive to noise and more stable than HD. AVD is simply the average HD over all pairs of points:

$$\mathrm{AVD}(G, S) = \max \left( \frac{1}{|G|} \sum_{x \in G} \min_{y \in S} \|x - y\|, \frac{1}{|S|} \sum_{x \in S} \min_{y \in G} \|x - y\| \right), \tag{11}$$

where $G$ and $S$ are the ground truth and the segmentation result respectively. $\| \cdot \|$ computes the Manhattan distance between the vector and origin. A smaller AVD indicates that there is more similarity between the prediction and the ground truth because AVD measures the volume difference in some sense.

Cohen's kappa coefficient (KAP) [101] is a statistical measure of inter-rater agreement of two samples, which is defined as

$$
\begin{aligned}
\mathrm{KAP} &= \frac{P_a - P_c}{1 - P_c} = \frac{f_a - f_c}{N - f_c} \\
P_a &= \frac{f_a}{N} \\
P_c &= \frac{f_c}{N} \\
f_a &= TP + TN \\
f_c &= \frac{(TN + FN)(TN + FP) + (FP + TP)(FN + TP)}{N},
\end{aligned}
\tag{12}
$$

where $P_a$ is the observed agreement between two samples and $P_c$ is the hypothetical probability of chance agreement. In our case, for example, for two volumetric segmentations, KAP can be expressed in terms of the corresponding frequencies $f_a$ and $f_c$, with $N$ denoting the number of voxels in the segmentation result. $f_a$ and $f_c$ can be further represented with regard to the four basic cardinalities of the confusion matrix, specifically, the true negatives (TN), the false negatives (FN), the true positives (TP) and the false positives (FP). KAP is a type of probabilistic metric that gives a higher KAP value for a more consistent segmentation result with the ground truth.

## Comparisons

Since there are several baseline methods for registration, atlas selection and label fusion, as well as our proposed method, we can choose various corresponding components to form different systems or pipelines. The average performance for each system is recorded in Table 3, with statistical significance test results for some top-performance systems shown in Table 4. All the segmentation evaluation results are averaged over all the target images in each dataset for each system. A series of detailed fusion results in slices for some picked samples are shown in Fig 6. In order to be succinct, we only demonstrate the results of the best fusion method for each registration method, for example, Joint Label Fusion results for ANTs SyN and Elastix without atlas selection in our case. The results indicate that our proposed pipeline, which consists of the integrated 3D flow-based registration, the energy-based atlas selection and the integrated 3D flow-based label transfer, achieves the best performance in terms of DC, KAP, AVD and computation time for segmenting one volume among all the competitive systems in all the datasets, with a statistically significant improvement ($p < 0.01$) over other systems. Atlas selection is a bonus for our proposed pipeline because it could rule out dissimilar atlas images which may not benefit the final label fusion result. However, atlas selection is beneficial to the fusion results of the label transfer and STAPLE but not to that of Joint Label Fusion according to Table 3.

## Influence of registration

We investigate the impact of various registration methods on performance by carrying out a complete registration experiment for each registration method on the five datasets and compute the Dice coefficient results of thousands of registration pairs (e.g., (33 AD + 71 MCI) × 47 NC × 2 = 9776 pairs for the ADNI dataset). The reason why we have this number of pairs is that we perform registration between each target volume and atlas volume, and we have built two sub-datasets for the left and right brain structure sub-volumes, respectively.

**Table 3. Final results on five datasets for systems with different methods in registration, atlas selection or label fusion modules.**

| Reg. | AS | Fusion | ADNI DC AD | ADNI DC MCI | ADNI AVD | ADNI KAP | ADNI Time | MICCAI 2012 DC | AVD | KAP | Time | LPBA40 DC | AVD | KAP | Time | Hammers DC | AVD | KAP | Time | IBSR-EADC DC | AVD | KAP | Time |
|---|---|---|---|---|---|---|---|---|---|---|---|---|---|---|---|---|---|---|---|---|---|---|---|
| ANTs | No | JLF | 0.859 | 0.865 | 0.184 | 0.861 | 328 | 0.790 | 0.401 | 0.786 | 80 | 0.795 | 0.369 | 0.791 | 240 | 0.812 | 0.231 | 0.811 | 165 | 0.767 | 0.330 | 0.765 | 867 |
| ANTs | No | STP | 0.758 | 0.779 | 0.313 | 0.770 | 327 | 0.758 | 0.416 | 0.754 | 80 | 0.770 | 0.431 | 0.764 | 239 | 0.813 | 0.224 | 0.812 | 164 | 0.766 | **0.313** | 0.763 | 857 |
| ANTs | No | LT | 0.848 | 0.851 | 0.187 | 0.848 | 327 | 0.751 | 0.463 | 0.747 | 80 | 0.769 | 0.446 | 0.764 | 239 | 0.803 | 0.247 | 0.802 | 164 | 0.245 | 2.572 | 0.242 | 857 |
| ANTs | Yes | JLF | 0.855 | 0.860 | 0.179 | 0.857 | 326 | 0.769 | 0.454 | 0.765 | 80 | 0.791 | 0.371 | 0.787 | 238 | 0.800 | 0.251 | 0.799 | 164 | 0.776 | 0.350 | 0.774 | 852 |
| ANTs | Yes | STP | 0.778 | 0.796 | 0.273 | 0.787 | 326 | 0.727 | 0.508 | 0.722 | 80 | 0.772 | 0.410 | 0.767 | 238 | 0.807 | 0.236 | 0.805 | 164 | 0.758 | 0.333 | 0.756 | 852 |
| ANTs | Yes | LT | 0.849 | 0.853 | 0.184 | 0.850 | 327 | 0.736 | 0.493 | 0.731 | 80 | 0.778 | 0.403 | 0.773 | 239 | 0.799 | 0.253 | 0.798 | 164 | 0.556 | 0.927 | 0.553 | 856 |
| Elastix | No | JLF | 0.852 | 0.859 | 0.194 | 0.855 | 85 | 0.778 | 0.442 | 0.774 | 27 | 0.751 | 0.465 | 0.745 | 58 | 0.797 | 0.254 | 0.796 | 50 | 0.763 | 0.334 | 0.760 | 265 |
| Elastix | No | STP | 0.730 | 0.748 | 0.346 | 0.739 | 84 | 0.714 | 0.507 | 0.708 | 27 | 0.752 | 0.479 | 0.746 | 57 | 0.804 | 0.239 | 0.802 | 50 | 0.746 | 0.348 | 0.743 | 256 |
| Elastix | No | LT | 0.798 | 0.807 | 0.252 | 0.802 | 84 | 0.723 | 0.515 | 0.719 | 27 | 0.675 | 0.784 | 0.668 | 57 | 0.765 | 0.308 | 0.763 | 50 | 0.288 | 2.823 | 0.285 | 256 |
| Elastix | Yes | JLF | 0.842 | 0.849 | 0.202 | 0.845 | 83 | 0.764 | 0.481 | 0.760 | 27 | 0.736 | 0.506 | 0.729 | 56 | 0.789 | 0.267 | 0.788 | 50 | 0.772 | 0.316 | 0.770 | 251 |
| Elastix | Yes | STP | 0.736 | 0.753 | 0.337 | 0.744 | 83 | 0.689 | 0.575 | 0.684 | 27 | 0.748 | 0.480 | 0.742 | 56 | 0.798 | 0.249 | 0.797 | 50 | 0.740 | 0.372 | 0.738 | 251 |
| Elastix | Yes | LT | 0.810 | 0.818 | 0.233 | 0.814 | 84 | 0.717 | 0.506 | 0.712 | 27 | 0.688 | 0.722 | 0.681 | 57 | 0.760 | 0.315 | 0.758 | 50 | 0.575 | 0.629 | 0.572 | 255 |
| ANTs | LTR | JLF | 0.852 | 0.857 | 0.186 | 0.854 | 201 | 0.790 | 0.362 | 0.786 | 91 | 0.789 | 0.387 | 0.784 | 156 | 0.815 | 0.226 | 0.814 | 133 | 0.724 | 0.407 | 0.721 | 251 |
| IF | No | JLF | 0.804 | 0.803 | 0.295 | 0.801 | 40 | 0.750 | 0.418 | 0.746 | 12 | 0.759 | 0.446 | 0.754 | 32 | 0.770 | 0.299 | 0.769 | 22 | 0.577 | 1.028 | 0.573 | 117 |
| IF | No | STP | 0.551 | 0.553 | 0.843 | 0.546 | 39 | 0.590 | 0.758 | 0.582 | 11 | 0.659 | 0.677 | 0.651 | 30 | 0.728 | 0.400 | 0.726 | 22 | 0.358 | 1.876 | 0.349 | 108 |
| IF | No | LT | 0.859 | 0.868 | 0.169 | 0.864 | 38 | 0.792 | 0.356 | 0.789 | 11 | 0.796 | 0.362 | 0.792 | 29 | 0.799 | 0.251 | 0.798 | 22 | 0.727 | 0.585 | 0.724 | 103 |
| IF | Yes | JLF | 0.793 | 0.794 | 0.302 | 0.791 | 38 | 0.726 | 0.462 | 0.722 | 11 | 0.759 | 0.439 | 0.753 | 30 | 0.779 | 0.284 | 0.778 | 22 | 0.586 | 0.924 | 0.583 | 102 |
| IF | Yes | STP | 0.587 | 0.588 | 0.729 | 0.582 | 38 | 0.561 | 0.794 | 0.553 | 11 | 0.661 | 0.652 | 0.652 | 29 | 0.730 | 0.390 | 0.728 | 22 | 0.393 | 1.584 | 0.386 | 102 |
| IF (ours) | Yes | LT | **0.867** | **0.875** | **0.156** | **0.871** | **37** | **0.802** | **0.321** | **0.798** | **11** | **0.809** | **0.317** | **0.805** | **29** | **0.829** | **0.206** | **0.828** | **22** | **0.778** | 0.363 | **0.775** | **102** |

Segmentation results are assessed with the Dice coefficient, the average Hausdorff distance in voxels and Cohen's kappa coefficient, while the execution time is measured in minutes. The average execution time for segmenting one volume over the whole test set is taken. Our methods and the best values are shown in bold. DC: the Dice coefficient. AVD: the average Hausdorff distance. KAP: Cohen's kappa coefficient. ANTs: ANTs SyN. IF: the integrated flow. AS: atlas selection. JLF: Joint Label Fusion. STP: STAPLE. LT: the label transfer. LTR: Learning to Rank. Reg.: registration.

**Table 4. Dice coefficient evaluation results of three registration methods and three best-performing systems on five datasets.**

| Method/System | ADNI | | | | MICCAI 2012 | | LPBA40 | | Hammers | | IBSR-EADC | |
|---|---|---|---|---|---|---|---|---|---|---|---|---|
| | AD | | MCI | | $DC(\mu \pm \sigma)$ | p-value | $DC(\mu \pm \sigma)$ | p-value | $DC(\mu \pm \sigma)$ | p-value | $DC(\mu \pm \sigma)$ | p-value |
| | $DC(\mu \pm \sigma)$ | p-value | $DC(\mu \pm \sigma)$ | p-value | | | | | | | | |
| ANTs | 0.691 ± 0.098 | $<10^{-30}$ | 0.706 ± 0.095 | $<10^{-30}$ | 0.646 ± 0.082 | $<10^{-30}$ | 0.656 ± 0.091 | $<10^{-30}$ | **0.735 ± 0.050** | $<10^{-30}$ | 0.605 ± 0.146 | <0.0001 |
| Elastix | 0.652 ± 0.084 | $<10^{-30}$ | 0.666 ± 0.081 | $<10^{-30}$ | 0.591 ± 0.078 | $<10^{-30}$ | 0.637 ± 0.089 | $<10^{-30}$ | 0.720 ± 0.052 | <0.0001 | 0.590 ± 0.133 | $<10^{-30}$ |
| **IF (ours)** | **0.775 ± 0.065** | - | **0.784 ± 0.064** | - | **0.714 ± 0.085** | - | **0.690 ± 0.078** | - | 0.724 ± 0.066 | - | **0.613 ± 0.126** | - |
| ANTs + JLF | 0.859 ± 0.049 | 0.0068 | 0.865 ± 0.049 | <0.0001 | 0.790 ± 0.095 | 0.13 | 0.795 ± 0.057 | <0.001 | 0.812 ± 0.038 | <0.00001 | 0.767 ± 0.037 | 0.028 |
| Elastix + JLF | 0.852 ± 0.046 | <0.0001 | 0.859 ± 0.047 | $<10^{-11}$ | 0.778 ± 0.095 | 0.021 | 0.751 ± 0.055 | $<10^{-21}$ | 0.797 ± 0.042 | $<10^{-9}$ | 0.763 ± 0.036 | 0.0029 |
| **IF + AS + LT (ours)** | **0.867 ± 0.038** | - | **0.875 ± 0.037** | - | **0.802 ± 0.077** | - | **0.809 ± 0.055** | - | **0.829 ± 0.037** | - | **0.778 ± 0.034** | - |

One-tailed paired t-tests with a significance level of 0.05 are performed for competitive methods against our method to test whether the mean difference between two sets of observations is significant. The best values are shown in bold. ANTs: ANTs SyN. IF: the integrated flow. JLF: Joint Label Fusion. LT: the label transfer. AS: atlas selection. DC: the Dice coefficient.

Some randomly selected registration results and three detailed voxel correspondence examples are graphically illustrated in Figs 7 and 8, respectively. Note that there are high-intensity contours in the results on LPBA40 with Elastix possibly because of preprocessing in Elastix but it has no overlap with the warped segmentation. A box plot of the registration results for all the pairs is shown in Fig 9, with statistical test results shown in the upper part of Table 4. It is clear that in our registration experiment, the integrated flow outperforms both ANTs SyN and Elastix by a statistically significant margin ($p < 0.0001$). The standard deviation of the integrated flow is smaller as well in most cases, which indicates its robustness.

We present precise displacement vectors for individual voxels upon registration in three picked examples from the ADNI and LPBA40 datasets in Fig 8. The bottom example demonstrates alignment between the hippocampus sub-volume of an MCI subject and that of a normal subject. The hippocampus in the target image is intensely deformed because the subject (i.e. the owner of the volume) is in the MCI stage, producing noticeable shrinkage of the hippocampus, and at the risk of developing Alzheimer's disease in the future. Such a pair of images requires a large non-linear deformation and a large search scale in the algorithm. The correspondences denoted by green and blue crosses in all three examples show that our integrated flow registration method performs well in detecting important landmarks on the outline of a brain structure for either small or large non-linear deformation. The large intensity-homogeneous region in the cuneus and the yellow crosses in the hippocampus suggest our good detection results in areas where low contrast appears.

## Influence of atlas selection

Between registration and fusion, we can perform atlas selection if we have some evaluation quantity that measures whether the warped atlas image is more similar to the target image, thus more likely to contribute to high segmentation accuracy.

We plot the Dice coefficient of the warped atlases as a function of their corresponding amount of deformation to study their underlying relationship. We choose the deformation amount instead of energies in order to make consistent and fair comparisons among registration approaches because all of them do not use the same energy definition. The deformation amount for registration in our case is defined as the sum of the Euclidean norms of all the displacement vectors. As shown in Fig 10, there is a roughly linear relationship between the deformation amount and the mean Dice, obviously observed in the second and third rows but not for the integrated flow in the first row. When registration induces a large amount of

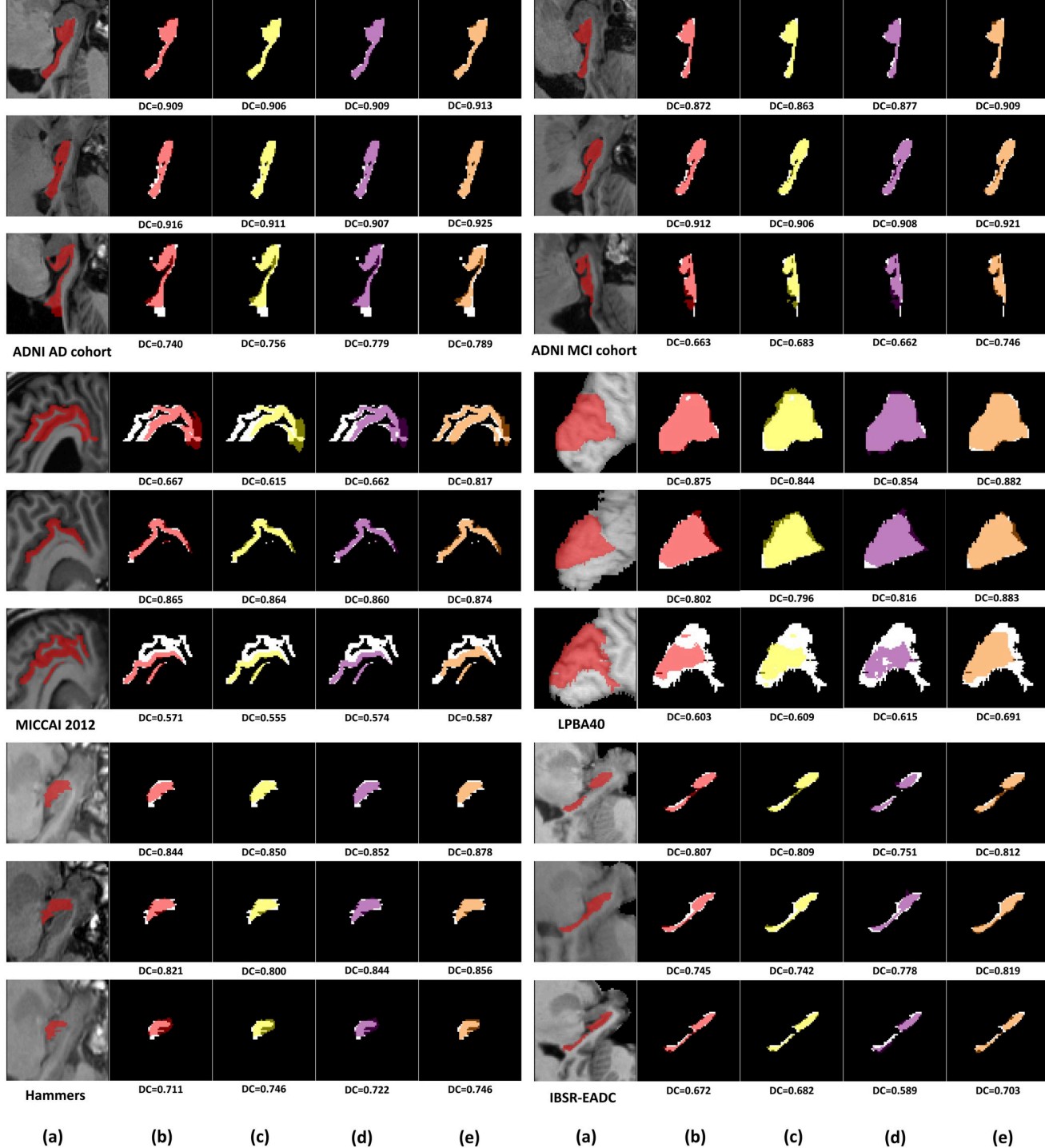

**Fig 6. Final fusion segmentation results of four best-performing systems.** Three target images are chosen for demonstration in each dataset or cohort. Cuneus sub-volumes in the LPBA40 dataset are shown as sagittal slices while all the other volumes are represented by coronal slices. (a) Target image and ground truth. (b) ANTs SyN + Joint Label Fusion. (c) Elastix + Joint Label Fusion. (d) ANTs SyN + Learning to Rank + Joint Label Fusion. (e) integrated flow + atlas selection + label transfer (our system). Ground truth segmentation is also shown in white color below each resulting segmentation in (b)-(e) for reference. DC: the Dice coefficient.

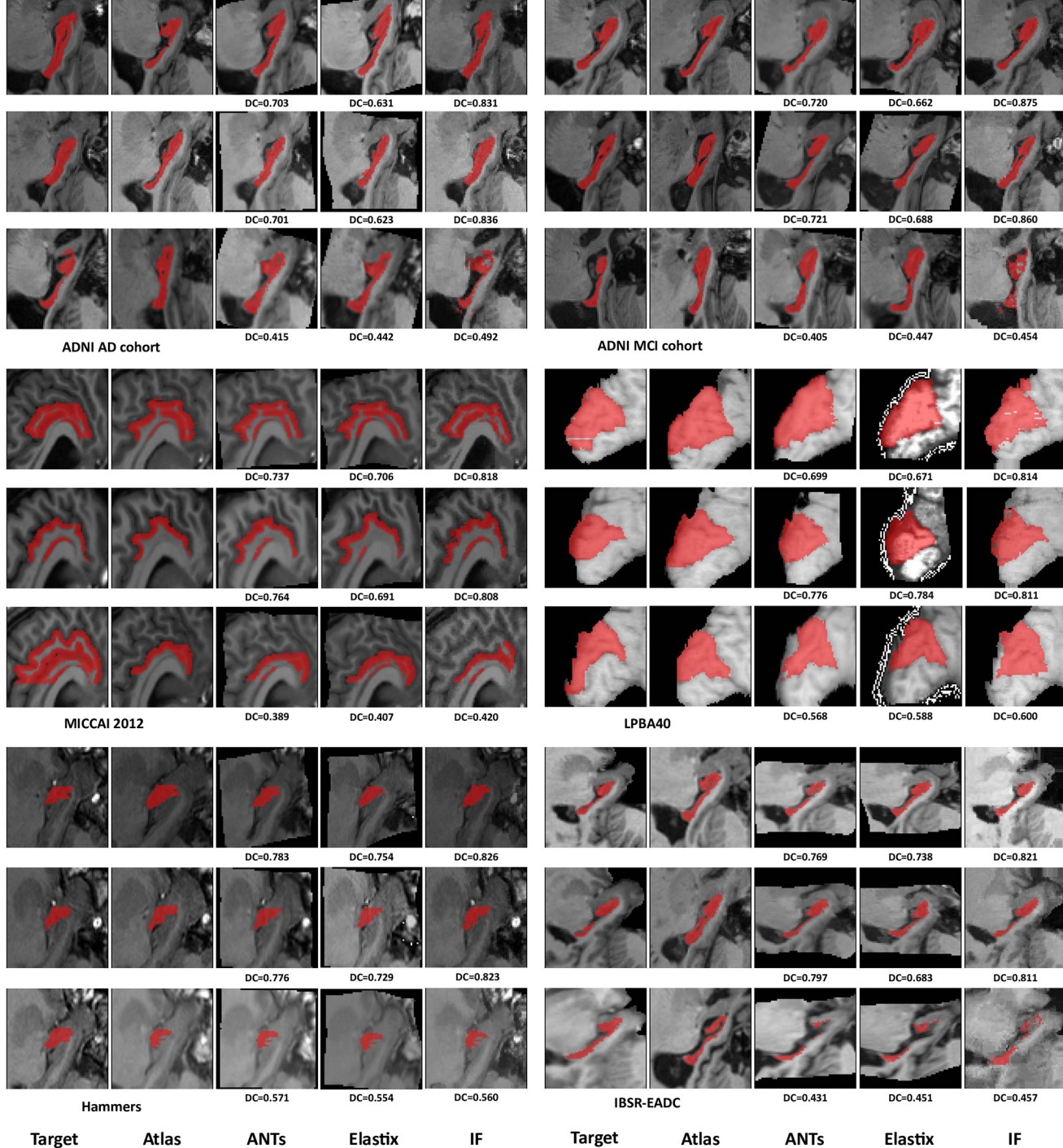

**Fig 7. Graphical comparisons of three registration methods.** Three examples from each dataset or cohort are illustrated. The shown images are the ground truth, the moving image, the registration result of ANTs, Elastix and our method, from left to right. Sagittal slices are adopted for the LPBA40 dataset while coronal slices are used for other datasets. ANTs: ANTs SyN. IF: the integrated flow (our method). DC: the Dice coefficient.

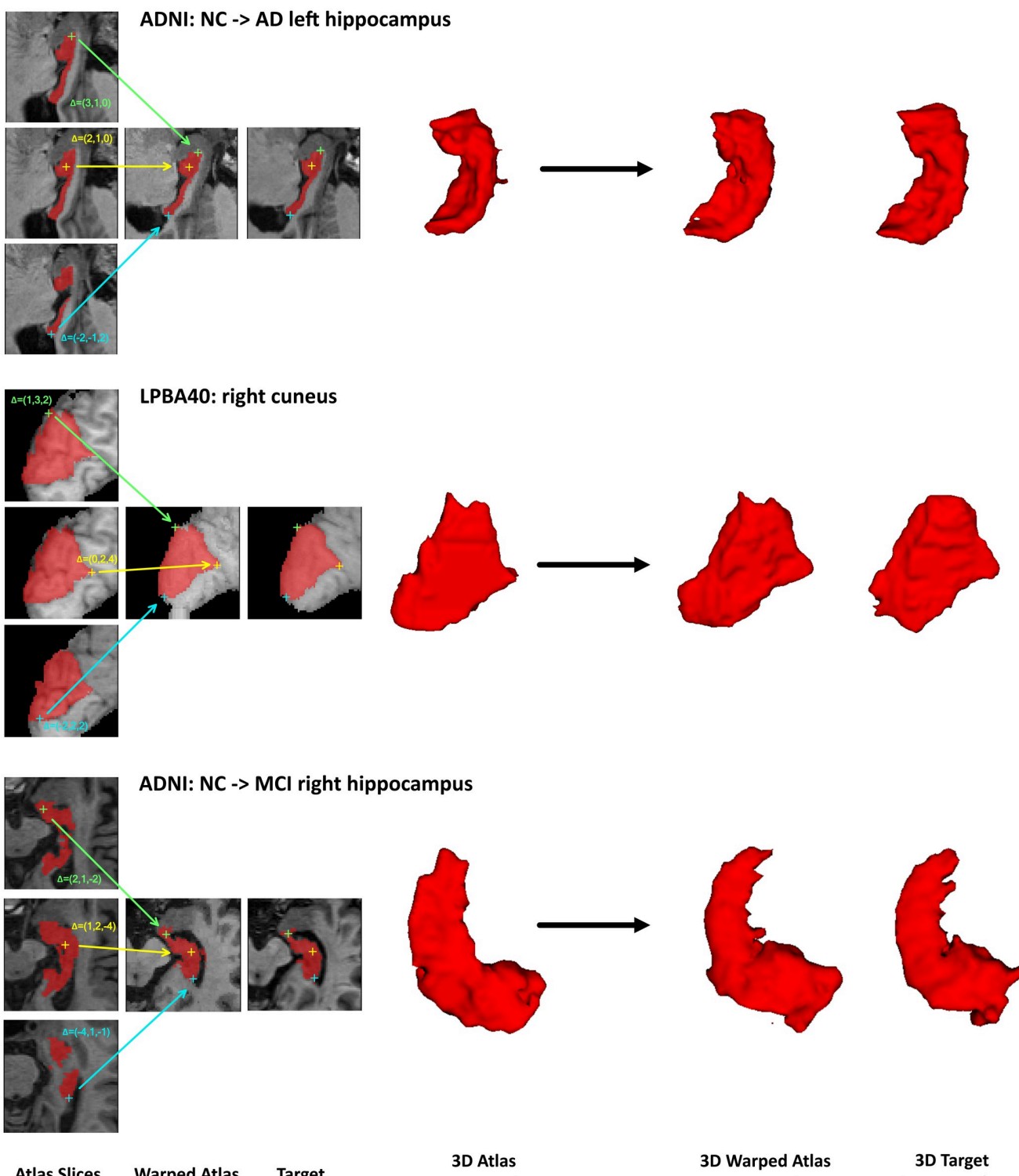

**Fig 8. Voxel correspondence of 3 pairs of subjects for our registration method.** The example in the top shows coronal slices of registration between an AD subject volume and an NC subject volume of the left hippocampus in the ADNI dataset. The sagittal slices in the middle demonstrate registration between two right cuneus volumes in the LPBA40 dataset. The registration example in the bottom shown in sagittal slices illustrates large deformation from an NC subject's right hippocampus volume to an MCI volume in the ADNI dataset. Three landmarks either on the boundary or in a homogeneous region for each example are annotated with crosses and displacement vectors in different colors. All the coordinates are in the native space of the target image.

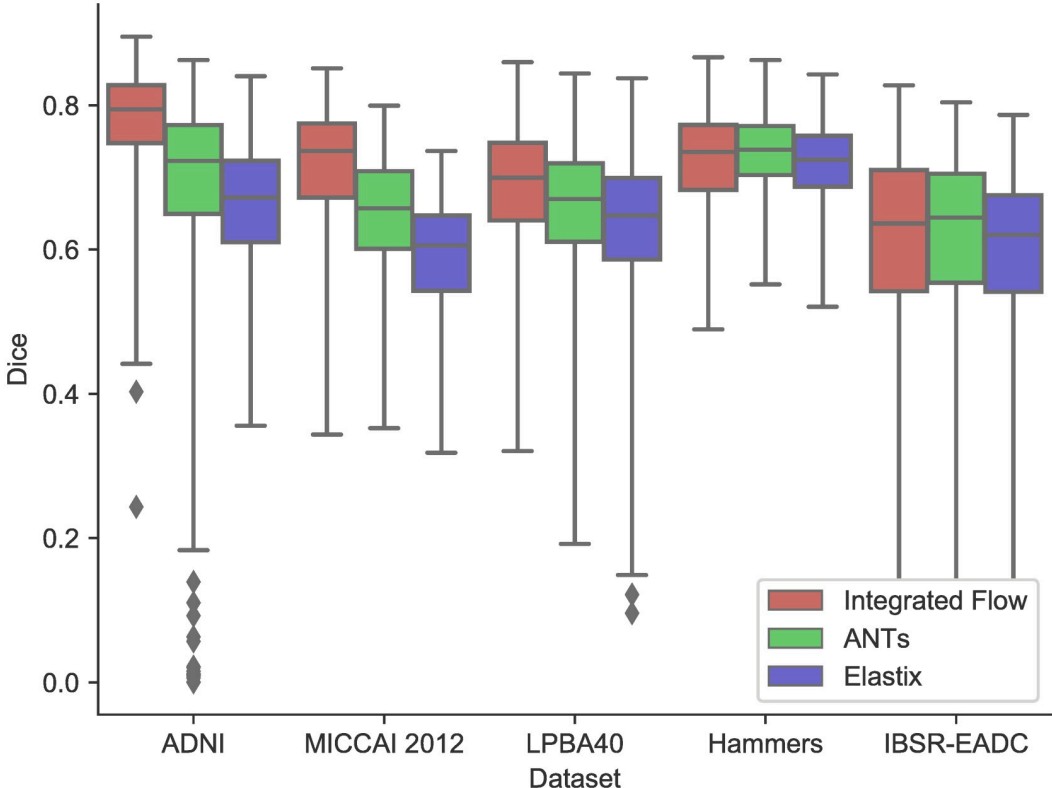

**Fig 9. A box and whisker diagram of registration results in Dice coefficients on five datasets for three registration methods, ANTs SyN, Elastix and the integrated flow (our method).**

deformation to transform the atlas, the resulting warped segmentation has a wide distribution of its Dice coefficients with regard to the ground truth, because even if the atlas image is dissimilar to the target image, there is still some chance to lead to a well-formed warped candidate atlas with high DC. In contrast, a small amount of transformation implies a similar atlas image to the target, which is more likely to result in a warped segmentation of good quality. Fig 10 also suggests that the approximate upper bound and lower bound of the DC distribution for each deformation amount are actually decreasing linearly as the deformation amount becomes larger. In other words, the expected Dice coefficient tends to become higher with less variance as the deformation amount decreases. It is noteworthy that we are only interested in the warped atlases with small deformation, whose expected quality is guaranteed. To conclude, deformation amount is a feasible atlas selection criterion with high confidence regarding atlas registration quality.

An experiment is conducted to study the relationship between the DC of the final segmentation and the number of selected candidates. To allow a fair comparison among three label fusion methods, we fix the registration method to be the integrated flow and perform atlas selection on the resulting warped atlas set with the optimization energy of the integrated flow as the selection criterion, while varying the number of candidate atlases. After sorting atlases according to their energies in ascending order, we choose the top $K$ candidates and warp them with their corresponding flow fields. Finally, Joint Label Fusion, STAPLE and the label transfer are applied to the same candidate atlas set for the final label fusion. The results of the atlas selection experiment on the ADNI dataset are plotted in Fig 11.

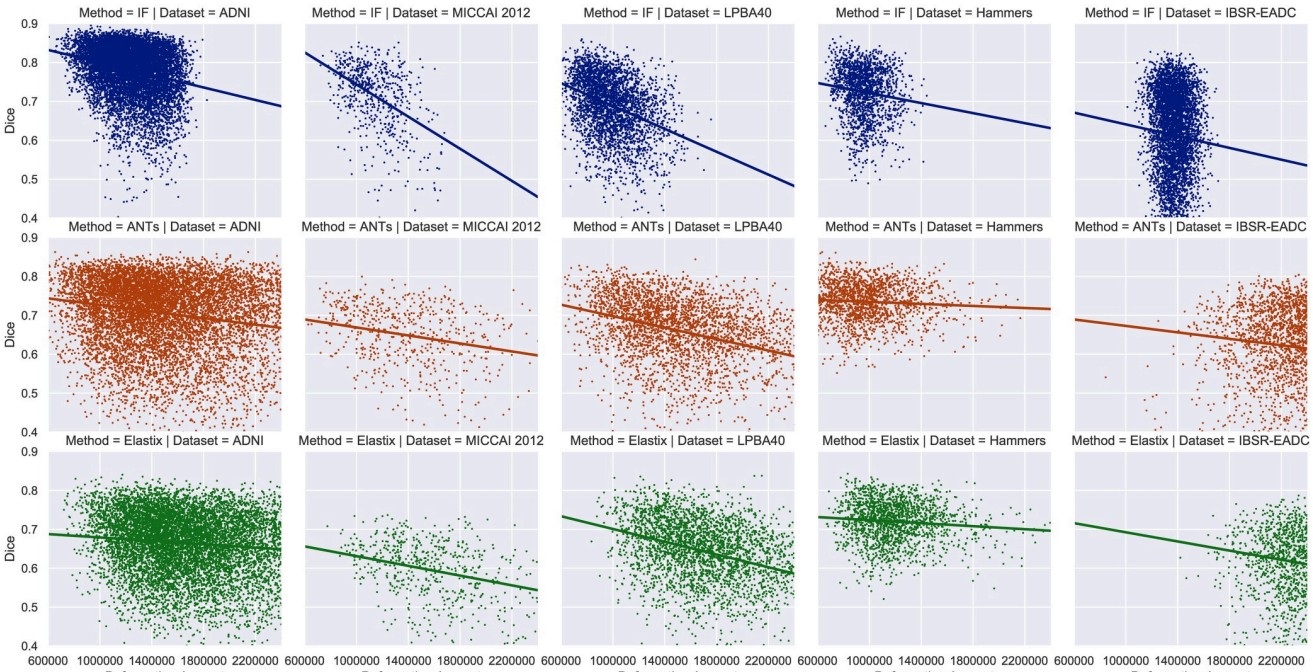

**Fig 10. Approximately linear relationships between the expected Dice coefficient and the amount of deformation in registration.** A subplot is made for each registration method and each dataset. The deformation amount is the sum of the Euclidean norms of all the displacement vectors. The best-fitting straight line through the data points is based on linear regression. For better illustration, some data points are out of range, thus absent in this figure. ANTs: ANTs SyN. IF: the integrated flow (our method).

We can infer from Fig 11 that there is an approximately optimal value for the atlas selection quantity. The three methods have different behaviors. In our method and STAPLE, the optimal segmentation accuracy is achieved at about 7-15 atlases, while Joint Label Fusion produces better segmentation as more atlases come in. Even when there is a large set of atlases, choosing several atlas candidates is sufficient for our proposed method to obtain an accurate segmentation, which is consistent with the results reported in [23, 27, 102–104].

## Execution time

We record the execution time of all the conducted experiments and compute the average time for individual modules as well as the whole system.

The average time taken to predict an annotated volume for one target image is shown in the Time column of Table 3. The results suggest that our proposed system is the most efficient one among all the competitive systems. For example, it takes a total of 11 minutes to generate segmentation for a target image in the MICCAI 2012 dataset, which is twice as fast as the Elastix system and at least 7 times faster than the ANTs system. It is also scalable and linear in the size of the atlas set by comparing the results across five datasets.

Registration time is the main contributor to the overall execution time. We compute the mean time for three registration methods with our datasets. As shown in Table 5, the integrated flow takes less than 1 minute for the registration of a pair of images while Elastix takes about 2 minutes and ANTs SyN spends about 6 minutes performing one registration. All the time values are measured in the same CPU compute node with a single-threaded limit for the sake of fairness.

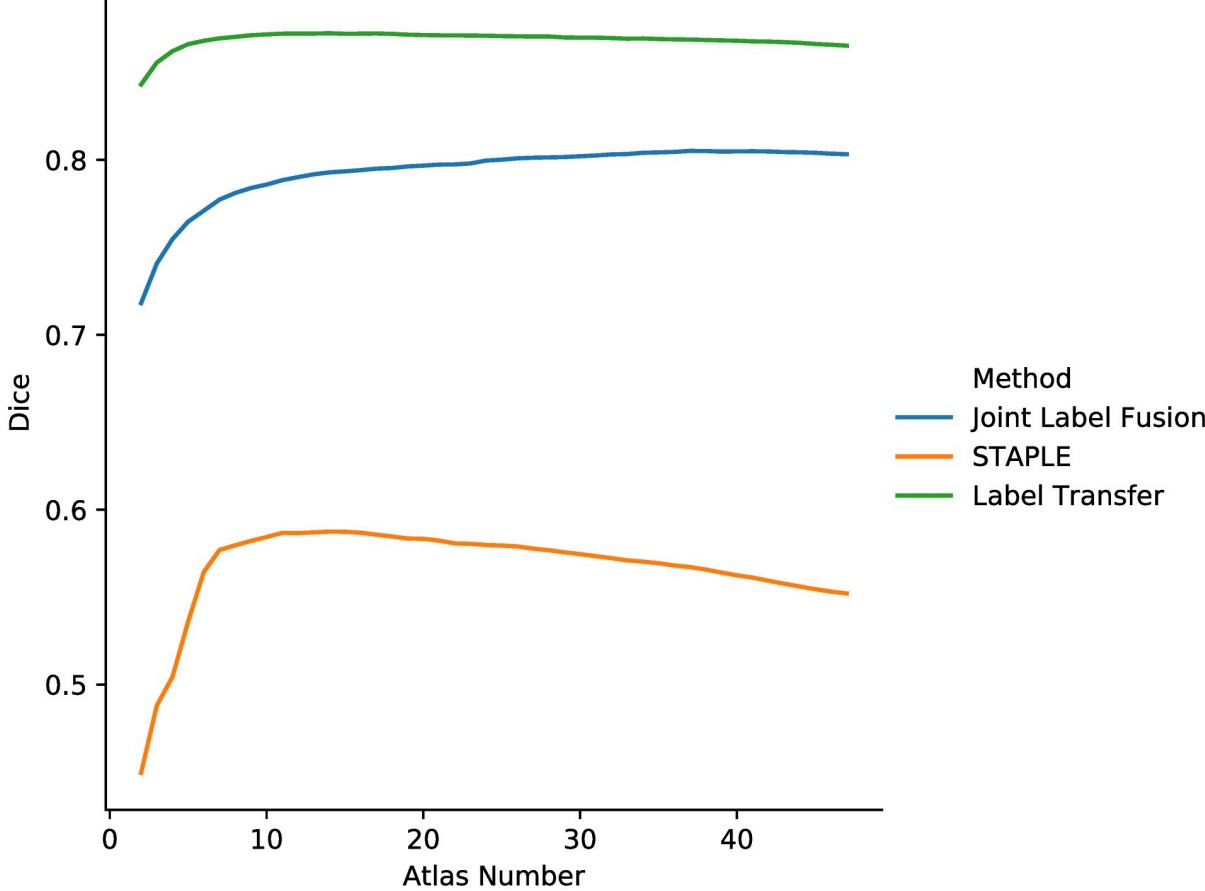

**Fig 11. The mean Dice coefficient results of three label fusion methods, Joint Label Fusion (middle), STAPLE (bottom) and the label transfer (top, our method), on the ADNI dataset.** The number of selected candidate atlases ranges from 1 to 47.

Although label fusion is not time-consuming as registration, it is still useful to record the time and observe the behaviors of the three adopted label fusion methods in this work, including Joint Label Fusion, STAPLE and the label transfer (our method). As shown in Fig 12, both the times of our proposed method and STAPLE are linear in the number of chosen candidates and grow slowly even when performing fusion for more than 100 atlases. STAPLE takes slightly more time than the label transfer does, but is significantly outperformed by it in segmentation results. In contrast, the time of Joint Label Fusion is approximately a quadratic function of the number of candidates, which empirically demonstrates that Joint Label Fusion takes into account the inter-dependency information among candidate atlases. The behavior is consistent with the computational analysis in [31]. Thus, STAPLE and our proposed method are more scalable than Joint Label Fusion.

**Table 5. Average execution time for three registration methods, the integrated flow (our method), ANTs SyN and Elastix.**

| Registration method | ANTs SyN | Elastix | Integrated flow |
|---|---|---|---|
| Average time | 339.483 | 102.931 | **45.319** |

The values are given in seconds.

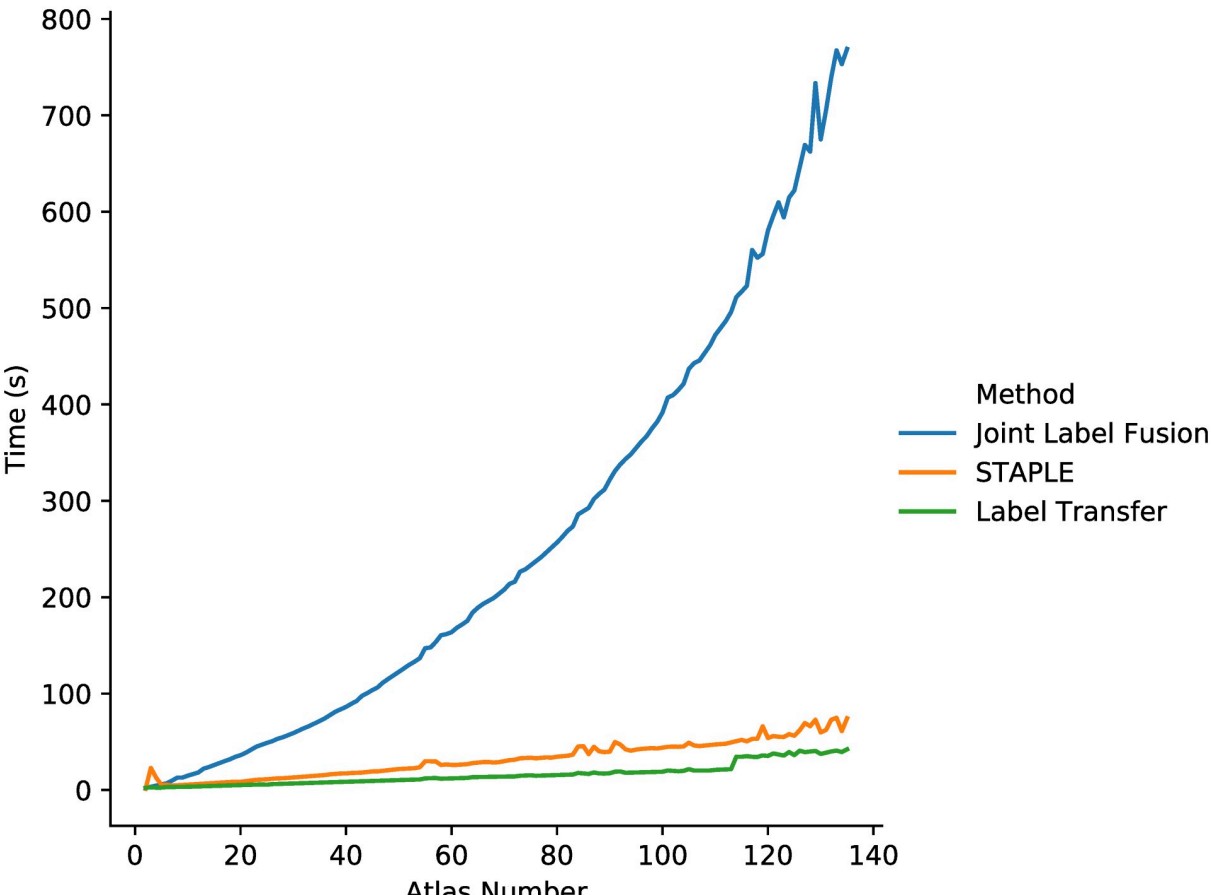

**Fig 12. Average execution time of label fusion programs with the number of atlas selection ranging from 1 to 135.** Three label fusion methods are Joint Label Fusion (top), STAPLE (middle) and the label transfer (bottom, our method).

To conclude, our proposed system, which consists of the integrated flow, the atlas selection and the label transfer, is much more efficient than other competitive methods in terms of both the whole system and individual modules.

## Disk space usage

We record the disk usage of three systems, our proposed system, ANTs SyN + Joint Label Fusion, and Elastix + Joint Label Fusion. Disk files include input dataset image files, output segmentation results and temporary files such as flow files and warped atlases.

Final segmentation files and input files are of the same size if converted to the same file format, so we only have to compare the temporary by-product files. According to Table 6, it is clear that the integrated flow requires less disk space than other methods upon registration. Usually, a system generates a transformation matrix, warps the atlas image with the matrix, and merges the warped candidate images and labels into a single label image file. So the warped volume files or flow files are the major part of a system's disk usage. In our method, we store only the flow files in the disk, after which we apply the flows to atlas images and labels at run time, without any warped image files generated to occupy disk space. We make our system more disk friendly by computing SIFT feature on the fly, which typically takes about 2 seconds in our datasets. In this way, our proposed system has the least disk usage among all the competitive systems.

**Table 6. Overall disk usage of three best-performing systems, measured in megabytes, based on our experiment on ADNI dataset.**

| Method | Raw | Feature | Reg. | Fusion | Total |
|---|---|---|---|---|---|
| **IF + AS + LT** | 679 | 0 | 7342 | 624 | 8645 |
| ANTs SyN + JLF | 679 | 0 | 136864 | 624 | 138167 |
| Elastix + JLF | 679 | 0 | 18477 | 624 | 19780 |

Our system is shown in bold. LT: the label transfer. IF: the integrated flow. JLF: Joint Label Fusion. Reg.: Registration. AS: atlas selection.

## Discussion

The hyper-parameters of the proposed methods are selected with grid search based on cross-validation results, which is standard practice for a machine learning task without an evaluation set. In addition to the guidance of cross-validation, the hyper-parameters should also be chosen to take into account memory limitation (searching window sizes) and tolerance for execution time (number of iterations, coarse-to-fine levels). During the tuning process, we observe that performance is sensitive to the hyper-parameters that depend on intensity distributions such as $\eta$ and $\alpha$, but insensitive to change in other integral values such as the number of hierarchies. Therefore, equipped with a standard protocol of preprocessing, there should be less effort in tuning the hyper-parameters dependent on the intensity statistics of the dataset.

Some failure cases when applying our methods are worth studying. The third lines of the examples in each dataset in Fig 7 present negative registration cases. One of the causes of such failure lies in the difference of modalities between the target and the atlas. For example, the negative case in the IBSR-EADC dataset makes the warped atlas image with our method less smooth than other positive cases. The large difference in modality and tissue structure forces ANTs SyN and Elastix to perform significant affine transformations, by inspecting the dark background voxels in the example slices, which further has an impact on the performance of their fine-grained deformable matching. The failure is attributed to the inherency of our algorithm as well. Because the integrated flow is a dense correspondence registration method that maps voxel to voxel instead of points of interest or supervoxel regions, it places great emphasis on voxel-wise global image similarity, which in turn might sacrifice some local segmentation consistency for better convergence and global performance. The negative examples in the ADNI AD and MCI cohort illustrate that our method produces a highly similar warped atlas image as the target but fails in some segmentation details. The third line in each dataset block in Fig 6 shows some of the negative fusion cases as Fig 7 does. Clearly, in these provided cases, all the systems make similar mistakes and have similar predictions in most of the regions. One explanation is that the atlas set is biased and in lack of variability. Failure in one registration pair does not have significant influence on the final fusion result if the atlas set is comprehensive and of good quality in a global perspective. However, in a biased low-quality atlas set, even if atlas selection is able to pick out few potentially good atlases after registration, the majority of the poor candidate atlases could dominate, which may result in a poor label fusion segmentation result.

In terms of the SIFT feature we adopt, for a three-dimensional image, the neighborhood and directions to choose for feature extraction are more complicated than for a two-dimensional image. In our method, only 6 directions and relatively small subblocks are taken into account when computing the feature vector. Extending the number of directions to 26 and enlarging the subblock size might be beneficial to better capturing the contextual information of a voxel, at the expense of a larger-dimensional feature vector.

**Table 7. Mean results of the incremental experiment with our system.**

| Metrics | +ADNI | +EADC AD | +EADC MCI | +EADC NC |
|---------|-------|----------|-----------|----------|
| DC | 0.838 | 0.858 | 0.865 | 0.871 |
| AVD | 0.190 | 0.164 | 0.156 | 0.147 |
| KAP | 0.836 | 0.857 | 0.863 | 0.870 |

From left to right in the top row, it represents the appending order of the datasets. DC: the Dice coefficient. AVD: average Hausdorff distance. KAP: Cohen's kappa coefficient.

Another thing about SIFT features is that it contributes to a large percentage of memory consumption since its dimension and data type require the operating system to allocate larger memory than other methods do. It is not feasible for a common personal computer even in 2D natural image applications as shown in [49], in which the experiments were conducted on a workstation with 32 GB memory. Henceforward, sacrificing data precision for lower memory consumption can be adopted though it may have little impact on segmentation accuracy.

To study the effectiveness of enlarging the atlas set, we carry out an additional experiment with our proposed system, in which the LMCI cohort volumes in the EADC dataset form the test set, while other volumes in the ADNI and EADC dataset are combined into the training set. Cropped hippocampus subvolumes are obtained in this experiment. Initially, the atlas set is empty. A complete multi-atlas segmentation experiment on the current target set and the current atlas set is performed as we append a set of additional atlases to the atlas set every time. We incrementally expand the atlas set by adding atlases to it in the following order: the ADNI dataset, the AD cohort volumes in the EADC dataset, the MCI cohort volumes in the EADC dataset and the NC cohort volumes in the EADC dataset. We report the average segmentation evaluation results in Table 7. It is shown that our system has learning ability and generates better predictions as more atlases come in.

We also record memory usage for registration and label fusion methods. We randomly select 5 training images and 1 test image from the MICCAI 2012 dataset and report the average maximum allocated memory for each method. The results are reported in Table 8. Methods that only make use of gray-scale features undoubtedly consume much less memory than our methods (IF and LT) do because a 48-dimensional SIFT feature vector is extracted for each voxel, expanding the size of an image by a factor of 48. So large memory usage is one of the disadvantages brought by the expressive SIFT feature.

In optimization of the integrated flow, a sequential BP approach is adopted in order to get better efficiency and convergence. In fact, it might be a good choice to replace sequential BP with a tree-reweighted message passing (TRW) approach to sacrifice some efficiency for better results when segmentation accuracy is the main concern. Moreover, the adopted min-sum belief propagation is in fact sensitive to the order of message updating. In contrast, an improved version of TRW called sequential TRW (TRW-S) [76] has a theoretical lower bound estimate non-decreasing guarantee and low memory consumption by some implementations.

**Table 8. Average maximum memory usage for three registration methods and two label fusion methods.**

| Method | ANTs | Elastix | IF | JLF | LT |
|--------|------|---------|-----|-----|-----|
| Average memory | 154.9 | **21.7** | 387.6 | 29.3 | 164.9 |

The values are given in megabytes. LT: the label transfer. IF: the integrated flow. JLF: Joint Label Fusion.

The final value of the energy function in the integrated 3D flow-based registration is adopted as the only standard for atlas selection. This is not always effective when intensity homogeneity is not guaranteed. The remaining noise or incomplete preprocessing may cause the elimination of a contributive candidate. Thus, there are improvements to be made on the parameters of the energy function and the structure of the formula. Moreover, some other similarity measures can be adopted and combined with the current energy function to achieve better performance for atlas selection.

Another thing about the energy-based atlas selection is that the decision is only made after all the registration is done, which is time-consuming if the training set is a large-scale atlas set even with fast registration. So instead of coming up with a more informative energy function, it would be good to filter out a subset of atlases with extremely fast algorithms before formal registration. The filtering algorithm could be coarse-grained but it really helps if the size of the given atlas set is extremely large, say, more than one thousand for a huge atlas database. The Learning to Rank method in [96] is motivating but the computation of SVM-Rank [83] is too expensive.

In the integrated 3D flow-based label transfer, the likelihood term is solely determined by the minimum difference of feature vectors which depends upon preprocessing and registration. To make it more robust, incorporating a voxel's contextual information into the likelihood term might be beneficial.

In our experiments, the obviously separable registration tasks are distributed to cluster computing nodes, which indeed reduces considerable time for a complete experiment from about several months to several hours. On top of this high-level parallelization, another consideration lies in the methods' internal structure parallelism. For instance, downsampling, window searching and optimizing are good places to experiment on to parallelize. Since we are not utilizing any GPU resources in the current implementation, an expected efficiency boost could be seen when we take advantage of the vector instructions.

Although it is an automated brain structure segmentation system that we propose, Freesurfer is included to preprocess the whole brain dataset to obtain cropped volumes for brain structures when we take into account the memory-intensive characteristics of the SIFT feature. Thus, it would be more practical if we can lower the memory demand in our implementation and make our system executable for a common personal computer on a whole brain dataset. Making improvements on this feature is expected to turn our method into a promising automated multi-atlas whole brain volume segmentation system. We leave it for future research work.

Deep learning models are promising and widely adopted non-linear hypothesis sets for statistical learning. Without sophisticated fine-tuning or a large hypothesis space, deep learning approaches are still not easily comparable to ANTs SyN [48]. The models for volumetric image segmentation usually require a long clock time to be trained even with GPU. Notwithstanding, it would be interesting to incorporate deep learning models in the registration, atlas selection and/or label fusion components, or even substitute the whole framework. If training time is not counted in registration time, its fast inference makes a significant difference. In datasets with limited training data, for example, MICCAI 2012, deep learning approaches are still not comparable to traditional methods such as ANTs SyN. For instance, according to the results reported in [46, 48], their proposed deep learning approaches are not comparable to traditional baselines with limited training data in classic settings and outperform them only with a much larger set of training data available. However, since neural network models form a larger hypothesis space and are able to learn task-specific representations automatically, we expect deep learning-based segmentation methods will benefit our multi-atlas segmentation

framework or outperform it completely in the form of end-to-end direct segmentation [105] even in the scarce-data setting.

## Conclusion

In this paper, we proposed a complete automated multi-atlas brain image segmentation system that consists of registration, atlas selection and label fusion, with an integrated flow connecting each target-atlas image pair. We developed an efficient energy-based atlas selection approach, a 3D coarse-to-fine flow matching scheme and a 3D ternary-layer message passing method for 3D sequential belief propagation. We conducted a series of extensive experiments on five publicly available datasets to compare our method with other methods. The results demonstrate that our method achieves comparable performance compared to some competitive atlas-based brain segmentation methods found in the literature in terms of computation time, accuracy, scalability and disk usage. The systematic pipeline is adaptable to image registration with large deformation. Anticipating that future work can focus on improvements of the model and applications of the system in fully automated whole brain segmentation, other medical image analysis and more general computer vision tasks.

## Acknowledgments

We would like to thank Gerard Sanroma, Hongzhi Wang and Alexander Shekhovtsov for all the helpful discussions.

## Author Contributions

**Conceptualization:** Yeshu Li, Yan Xu.

**Data curation:** Yeshu Li, Ziming Qiu, Xingyu Fan, Eric I-Chao Chang.

**Formal analysis:** Yeshu Li, Yan Xu.

**Investigation:** Yeshu Li, Ziming Qiu.

**Methodology:** Yeshu Li, Yan Xu.

**Project administration:** Yeshu Li, Xianglong Liu, Eric I-Chao Chang, Yan Xu.

**Resources:** Yeshu Li, Ziming Qiu, Yan Xu.

**Software:** Yeshu Li, Ziming Qiu.

**Supervision:** Xianglong Liu, Eric I-Chao Chang, Yan Xu.

**Validation:** Yeshu Li, Ziming Qiu.

**Visualization:** Yeshu Li, Ziming Qiu, Xingyu Fan.

**Writing – original draft:** Yeshu Li, Yan Xu.

**Writing – review & editing:** Yeshu Li, Ziming Qiu, Xingyu Fan, Yan Xu.

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
