## [Decision Letter · Decision Letter 0]

7 Jun 2021

PONE-D-21-07784

Integrated 3d flow-based multi-atlas brain structure segmentation

PLOS ONE

Dear Dr. Xu,

Thank you for submitting your manuscript to PLOS ONE. After careful consideration, we feel that it has merit but does not fully meet PLOS ONE’s publication criteria as it currently stands. Therefore, we invite you to submit a revised version of the manuscript that addresses the points raised during the review process.

Based on the comments received from the reviewers and my own observation I recommend major revisions for the article.

We look forward to receiving your revised manuscript.

Kind regards,

Thippa Reddy Gadekallu

Academic Editor

PLOS ONE

Journal Requirements:

'YX was supported by the National Science and Technology Major Project of the Ministry of Science and Technology in China under Grant 2017YFC0110903, the National Natural Science Foundation in China under Grant 62022010, 81771910, SinoUnion Healthcare Inc. under the eHealth program, the Fundamental Research Funds for the Central Universities of China from the State Key Laboratory of Software Development Environment in Beihang University in China, the 111 Project in China under Grant B13003, the high performance computing (HPC) resources at Beihang University. The funders had no role in study design, data collection and analysis, decision to publish, or preparation of the manuscript.'

We note that one or more of the authors are employed by a commercial company: Microsoft Research, Beijing

We also note that you received funding from a commercial source: SinoUnion Healthcare Inc.

b. Please also provide an updated Competing Interests Statement declaring the commercial affiliation and explicitly stating the commercial funder, along with any other relevant declarations relating to employment, consultancy, patents, products in development, or marketed products, etc.  

Reviewers' comments:

Reviewer's Responses to Questions

**Comments to the Author**

1. Is the manuscript technically sound, and do the data support the conclusions?

Reviewer #1: Yes

Reviewer #2: Yes

2. Has the statistical analysis been performed appropriately and rigorously? 

Reviewer #1: Yes

Reviewer #2: Yes

3. Have the authors made all data underlying the findings in their manuscript fully available?

Reviewer #1: Yes

Reviewer #2: Yes

4. Is the manuscript presented in an intelligible fashion and written in standard English?

Reviewer #1: Yes

Reviewer #2: Yes

5. Review Comments to the Author

Reviewer #1: This paper proposes a new brain ROI labeling method, which belongs to the multi-atlas registration and label fusion framework. Specifically, it uses the 3D vector flow to align the atlas onto the segmenting image, then use the energy generated during the alignment for atlas selection. After that, the labels from different atlases are fused together using a flow-based label transfer to form the final segmentation. The paper uses three datasets to validate their method achieves a superior performance. The entire method is sound, and the overall idea is clear. The reviewer has the following major and minor concerns.

Major concerns:

1. The SIFT flow works well on the 3D liver image segmentation, however, the brain is different from the liver in its convoluted structures. Aligning across different brain is much more challenging since different subjects may have substantial different cortex in folding, which means the alignment can cause large local deformations. This paper introduces the vector flow to do the alignment, but the smooth term in equation (3) seems not to be enough to guarantee the deformation to be smooth.

2. To solve the optimization of the 3D vector flow, the paper uses a coarse-to-fine strategy to get the optimal solution. However, in the label fusion stage, the optimization seems similar, is it beneficial to use the coarse-to-fine strategy?

3. Is there any specific reason to choose grayscale value as additional features? It is interesting to know whether introducing it is beneficial to the segmentation performance. Also, is \\zeta a weight factor for gray scale feature? It is not clear how coefficient \\zeta works in the feature concatenation.

4. From the paper, the author mentions their registration outperforms the ANTs. Since ANTs is a relatively old registration method, latest registration like VoxelMorph seems outperforms the ANTs, can the author discuss the use of the deep learning-based registration in the proposed framework?

5. Since the end-to-end segmentation using deep learning has achieved very good performance in the segmentation, the latest segmentation network like U-net and Dense-net also achieves very good performance in the brain ROI segmentation, at least, the author needs to discuss the comparison with deep learning-based methods. In the paper, the author mentioned the major limitation for the deep learning method is the need of manual segmentation and computation cost, but for the patch based deep learning method, the require of the manual segmentation can be reduced. Also, the paper mainly discussed the computation cost in CPU, while use GPU is very common in many applications. Would there be any limitation of using GPU?

6. Minor comments, it is very strange the figures are separated with the figure captions.

Reviewer #2: 1. The paper is written in a good manner. Some minor touches can improve this paper more.

2. The quality of the figures can be improved more.

3. The contributions of the authors are not clear. They have mentioned in first contribution.

4. Future research directions section is core, however, it is not good at all.

5. What are the computational resources reported in the state of the art for the same purpose?

- Please cite each equation and clearly explain its terms.

- Clearly highlight the terms used in the algorithm and explain them in the text.

6. What are the evaluations used for the verification of results?

7. Several paragraphs contain trivial information and should be dropped.

8. I found some English mistakes please check them.

9. Kindly refer the below paper:

1. Rajput, D.S., Basha, S.M., Xin, Q. et al. Providing diagnosis on diabetes using cloud computing environment to the people living in rural areas of India. J Ambient Intell Human Comput (2021). https://doi.org/10.1007/s12652-021-03154-4

6. PLOS authors have the option to publish the peer review history of their article (what does this mean?). If published, this will include your full peer review and any attached files.

Reviewer #1: No

Reviewer #2: No

---

## [Author Response · Author response to Decision Letter 0]

23 Jul 2021

The detailed responses are attached as a pdf file.

---

## [Decision Letter · Decision Letter 1]

16 Mar 2022

PONE-D-21-07784R1Integrated 3d flow-based multi-atlas brain structure segmentationPLOS ONE

Dear Dr. Xu,

Thank you for submitting your manuscript to PLOS ONE. After careful consideration, we feel that it has merit but does not fully meet PLOS ONE’s publication criteria as it currently stands. Therefore, we invite you to submit a revised version of the manuscript that addresses the points raised during the review process.

The revised version of manuscript has been reviewed 2 reviewers. The reviewers were of the view that authors have done revisions to improve the manuscript, however, some of the reviewer issues seems not fully addressed. After considering the comments of both reviewers, my decision is "major revision". Please incorporate the suggestions/comments of both reviewers. 

We look forward to receiving your revised manuscript.

Kind regards,

Gulistan Raja

Academic Editor

PLOS ONE

Reviewers' comments:

Reviewer's Responses to Questions

**Comments to the Author**

1. If the authors have adequately addressed your comments raised in a previous round of review and you feel that this manuscript is now acceptable for publication, you may indicate that here to bypass the “Comments to the Author” section, enter your conflict of interest statement in the “Confidential to Editor” section, and submit your "Accept" recommendation.

Reviewer #1: (No Response)

Reviewer #3: (No Response)

2. Is the manuscript technically sound, and do the data support the conclusions?

Reviewer #1: Partly

Reviewer #3: Yes

3. Has the statistical analysis been performed appropriately and rigorously? 

Reviewer #1: Yes

Reviewer #3: Yes

4. Have the authors made all data underlying the findings in their manuscript fully available?

Reviewer #1: (No Response)

Reviewer #3: Yes

5. Is the manuscript presented in an intelligible fashion and written in standard English?

Reviewer #1: Yes

Reviewer #3: Yes

6. Review Comments to the Author

Reviewer #1: Although the authors have done some revisions to improve the manuscript, some of the reviewer issues seems not fully addressed.

1. For the smoothness issue, the reviewer means for neighboring voxels, their displacement may have relatively large differences. This can be easily inferred from eq. (3), for the voxels with larger displacement field, the neighborhood displacement would have no effect, because it will automatically choose the lower bound, i.e., d. This means when doing the optimization, the neighborhood regularization term may not contribute to the objective function if the previous L1 norm cannot limit the displacement vector to be sufficient small. For the brain registrations, it is agreed for some region, the displacement can be large due to the substantial structure difference across individuals. At those regions, would the neighborhood smoothness term fail and cause the displacement field non-smooth? It is better to show some displacement field visualization figures to illustrate this?

2. For the comparison to the deep learning-based method, is there any reference or experiments support the expression “In datasets with limited training data, for example, MICCAI 2012, deep learning approaches are still not comparable to traditional methods such as ANTs SyN.” in the paper. The current clarification seems not convincing.

Reviewer #3: In my opinion the paper is well written and structured and conduct a significant number of experiments to support the authors' conclusions. However, I have found some issues the authors should address before the final acceptance of the manuscript. They are as follows:

1.- Concerning eq. 1 the authors determine the magnitude and two orientations for the gradient in 3D. The magnitude is clearly defined. For the orientations two orientations are defined, theta and phi. The authors should clearly explain, perhaps using a figure, the exact meaning of the two orientations. Are they those of spherical coordinates?

2.- In the paragraph after eq. 1 (lines 180-185), a neighborhood of a voxel is defined by an 8x8x8 cube. Then the authors said that the concerning voxel is in the center. How is this possible if the size of the neighborhood is even (8x8x8)? Why not to have an odd number for the neighborhood size, such as 9? In this case the neighborhood will be of size 9x9x9, being the center voxel clearly defined. Using an odd number for the size, how are the sub-blocks defined?

3.- A gaussian weighting function is applied for the sub-block voxels (lines 188-). This could be done in several ways. Pleas add an equation for that part. A gaussian function has the sigma parameter to be adjusted. How is this selected? Please add a table with the 6 histogram bins, giving the specific theta and phi intervals (the authors just give a particular case in line 191). I do not understand the statement: "Such division is not disjoint thus introducing overlap among bins because this is not the geographical way of creating parallels and meridians". Same for "o overcome rotation dependence, we subtract the dominant orientation of the center voxel from all the orientations so that the dominant one points to (θ = 0, φ = 0) and rotation invariance is thus guaranteed." Please rewrite those expressions with some more details.

4.- Caption of figure 3 is talking about left-right, but figure 3 has two diagrams in the bottom-top position.

5.- Concerning eq. 3, please elaborate a bit more on the need of t and d parameters. Should they be large? Why are they needed? Comparing this eq. with eq. 6, and for the sake of homogeneity, authors could add the "+ log Z" and the end of eq. 3 (as done in eq. 6). Indeed, in the MRF context, this Z is normally referred to as "partition function". A MRF is indeed a probability field which is easy to determine out of the Energy (the E function as defined for instance in eq. 3 or the F function as defined in eq. 6). For the sake of completeness, could the authors elaborate a little bit on that? For instance, adding some equations concerning the probabilities. This way is easy to see that the achieved global minimum energy corresponds to the global maximum of the probabilistic field, i.e., the MAP solution.

6.- In line 262, please remove the "1" at the end of the author's name of [79]: "Shekhovtsov1" -> "Shekhovtsov".

7.- Is possible to add a reference for paragraph in lines 305-308? In particular, the paper reads: "Hence it is not feasible in practice, especially in 3D medical image analysis." Add a reference for this or an explanation with some evidence for that fact.

8.- In paragraph from line 309-, the original image is downsample several times. It is well known that downsampling without pre-filtering causes aliasing. Is that fact taken into account when downsampling the images? The correct way to downsample for a 2 factor is to filter the input using a low-pass filter with half the original bandwidth and then perform the downsample (for instance keeping one sample each two). Please, elaborate a bit more on this on the paper.

9.- In lines 363-, the label transfer method is explained. The notation is not consistent with respect to that used in algorithm 1 (page 8 of the manuscript). In particular, in algorithm 1, atlas labels are denoted by L and L_i, and target labels with L_target, however, in lines 363- (page 16), the notation is different, c and c_i are used for the labels: c_i for the atlas and c for the target. Is possible, for instance, to used L_i and L_target?

10.- In lines 397-400, the manuscript reads: "It is noteworthy that in comparison to applications in natural scene images, medical image registration and label transfer applications require the coefficients of the smoothness and spatial term to be much higher because of the intensity characteristics in medical images." Please add a reference to the literature or some more evidence for this to happen. How are the "intensity characteristics in medical images"? Is that due to the fact that in medical images, the shapes are smoother (rounded)?

11.- In line 402, what is the meaning for the "D" parameter? Are you considering color images or so? What is the D value for the images used in the experiments? Is it possible to consider volumes with different dimensions? Authors said NxNxN, but this is not always true: usually the volume is NxNxM, MxNxN or NxMxN, depending of the acquisition plane (axial, sagittal, coronal).

12.- In caption of table 1, please add the meaning of the following acronyms: AC, MCI, NC and LMCI. I think they are defined in the main text, but for easy reading, I think they should be defined in the caption as well.

13.- Concerning fig.4, is possible to merge last two rows (on the left): the two ROIs (left and right) as in the other rows? In this figure, row 4 (LPBA40), column 1, I cannot see anything in the right ROI.

14.- In lines 558-559: "In integrated 3D flow-based registration, t is computed based on the current data histogram normalization thus is data-dependent", please explain. How is that performed? Which value/s is/are used for the experiments?

15.- I think the acronym BP-S is not defined. Please define it first time.

16.- In lines 563 and 564, indicate that the given value refers to the K parameter.

17.- In lines 581 and 582, AVD is defined as Average Hausdorff Distance, but it seems more natural to use AHD as the acronym, why AVD? In addition to that, it happens that in eq. 11 a maximum operator is used to mix up the two possible Hausdorff distances. I do not understand why to say "average" as a maximum is used. Should you change eq. 1 using a semi-sum?

18.- In light of eq. 12, please add the following definitions (if true): Pa=fa/N and Pc=fc/N.

19. Line 611 and table 2: please make it clear that the execution time is for each of the test cases, not for whole test set. This point does not seem clear.

20.- Fig. 5 and its caption: column (d) is for Learning to Rank atlas selection, but which registration algorithm is used? Any fusion step? Which one? Please make this point clear in the caption of that figure.

21.- Fig. 6. There is an artifact (false contour which possible saturation of the intensity) for the LPBA40 and Elastix case images. Please fix or explain the reason for that.

22.- Concerning results of fig. 9, the authors said that there is a linear relationship between the two considered variables. Two is fine for the second and third rows. However, for the proposed method (row 1), I do not see a clear linear relationship. In fact, the linear regression does not explain much of the cloud of points. Please, take this fact into account and modify your conclusions accordingly.

23.- In experiment concerning fig. 9, the Dice coefficient is represented as a function of the corresponding amount of deformation. However, the atlas selection is done using the final energy of registration part (using the MRF formulation). Why not to compare the Dice to the energy or deformation to energy, for instance? This will justify the part given in lines 666- (page 29). For instance, a figure similar fig. 10, but showing the energy and atlas number after the sorting.

24.- A section showing the "Disk Space Usage" is presented. However, I think the "memory need" for the algorithms to run is nowadays even more important than the "disk usage". Is possible to add a table similar to table 5, with the memory used? Add a paragraph with a discussion about the achieved results in the main text. Some mention is given in the discussion section (line 768), but in my opinion, a table should be included with memory usage and done for the disk.

25.- Concerning the meta-parameters of the proposed algorithm (t, eta, alpha, etc.), in lines 557- (page 23) some explanation and values used are provided. Please add a paragraph in the "Discussion" about this point: how to select these parameters for a given scenario, how sensitive will the results be with respect to the selection of these meta-parameters, and so forth.

26.- Last paragraph in the Discussion deals with Deep Learning (DL) and Artificial Intelligence (AI). However, I see three places DL/AI algorithms can be applied in the context of the paper: in the registration part, in the atlas selection part and in the fusion part. DL/IA can also be applied as a whole: as a pure segmentation algorithm. In my opinion, the authors should elaborate more about that point.

27.- In lines 849-850, the manuscript reads: "The results demonstrate that our method achieves comparable performance compared to some competitive brain segmentation tools in terms of computation time, accuracy, scalability and disk usage.", I cannot see evidence of that fact. Authors compare their algorithm with respect to other atlas-based segmentation algorithms. However, no comparison is performed with respect to other segmentation algorithms in general. Paper can conclude that the proposed method outperforms other atlas-based segmentation found in the literature (in execution time, accuracy, etc.). Please justify or change that statement.

28.- I do not see the need to include ref. 105: in my opinion, it is an out-of-scope work: diabetes has nothing to do with the content of the paper.

7. PLOS authors have the option to publish the peer review history of their article (what does this mean?). If published, this will include your full peer review and any attached files.

Reviewer #1: No

Reviewer #3: No

---

## [Author Response · Author response to Decision Letter 1]

30 Apr 2022

See the uploaded file for the response to reviewers.

---

## [Decision Letter · Decision Letter 2]

2 Jun 2022

PONE-D-21-07784R2Integrated 3d flow-based multi-atlas brain structure segmentationPLOS ONE

Dear Dr. Xu,

Thank you for submitting your manuscript to PLOS ONE. After careful consideration, we feel that it has merit but does not fully meet PLOS ONE’s publication criteria as it currently stands. Therefore, we invite you to submit a revised version of the manuscript that addresses the points raised during the review process.

The revised manuscript had been reviewed by 2 reviewers. Both reviewers had made some minor suggestions to further improve your work and recommended 'minor revision'. After consideration of comments of both reviewers, my decision is 'minor revision'. Please incorporate the comment/suggestions of both reviewers. 

We look forward to receiving your revised manuscript.

Kind regards,

Gulistan Raja

Academic Editor

PLOS ONE

Journal Requirements:

Reviewers' comments:

Reviewer's Responses to Questions

**Comments to the Author**

1. If the authors have adequately addressed your comments raised in a previous round of review and you feel that this manuscript is now acceptable for publication, you may indicate that here to bypass the “Comments to the Author” section, enter your conflict of interest statement in the “Confidential to Editor” section, and submit your "Accept" recommendation.

Reviewer #1: (No Response)

Reviewer #3: All comments have been addressed

2. Is the manuscript technically sound, and do the data support the conclusions?

Reviewer #1: Yes

Reviewer #3: Yes

3. Has the statistical analysis been performed appropriately and rigorously? 

Reviewer #1: N/A

Reviewer #3: Yes

4. Have the authors made all data underlying the findings in their manuscript fully available?

Reviewer #1: Yes

Reviewer #3: Yes

5. Is the manuscript presented in an intelligible fashion and written in standard English?

Reviewer #1: Yes

Reviewer #3: Yes

6. Review Comments to the Author

Reviewer #1: For the question of the smoothness, the author response that the discontinuity is actually desirable in applications like brain image registration, while the reviewer has a different view on this. Most latest brain registration can theoretically preserve the smoothness (like Demons, ANTs, Voxel Morph etc.). Although practically, some discontinuity may rise due to the discretization. The smoothness is also important for accurate alignment, especially when the deformation is large. While the proposed method seems can not preserve the smoothness in theory. In Fig. 4, the author visualized the deformation field with stronger smoothness generated with integrated flow. So is that means in practical, the proposed method can generate the smooth deformation field in most cases?

Reviewer #3: Many thanks for the changes made on the paper. Now in my opinion the paper reads better.

There are still some minor issues that authors should change:

1.- In algorithm 1 item number 2: c_target should be L_target (see item number 17 in that algorithm).

2.- Line 396 c_i should be L_i (see line 395).

3.- Line 398, the paper reads: "We aim to use the above information to generate c", c should be changed by L. Please, check the notation coherence along the paper concerning the labels (c vs L).

4.- Line 406 reads: "The first sigma summand term", please remove the word "sigma" or rewrite it as: The first summation, or The first term in that equation.

5.- There are other minor typos. I recommend to proofread the whole paper to fix them.

7. PLOS authors have the option to publish the peer review history of their article (what does this mean?). If published, this will include your full peer review and any attached files.

Reviewer #1: No

Reviewer #3: No

---

## [Author Response · Author response to Decision Letter 2]

7 Jun 2022

The response to reviewers is uploaded as a document.

---

## [Editor Report · Decision Letter 3]

9 Jun 2022

Integrated 3d flow-based multi-atlas brain structure segmentation

PONE-D-21-07784R3

Dear Dr. Xu,

We’re pleased to inform you that your manuscript has been judged scientifically suitable for publication and will be formally accepted for publication once it meets all outstanding technical requirements.

Kind regards,

Gulistan Raja

Academic Editor

PLOS ONE
---

## [Editor Report · Acceptance letter]

5 Aug 2022

PONE-D-21-07784R3 

Integrated 3d flow-based multi-atlas brain structure segmentation 

Dear Dr. Xu:

I'm pleased to inform you that your manuscript has been deemed suitable for publication in PLOS ONE. Congratulations! Your manuscript is now with our production department. 

Kind regards, 

on behalf of

Dr. Gulistan Raja 

Academic Editor

PLOS ONE